# How Neural Networks Extrapolate: From Feedforward to Graph Neural Networks

**Keyulu Xu**[†]**, Mozhi Zhang**[‡]**, Jingling Li**[‡]**, Simon S. Du**[§]**, Ken-ichi Kawarabayashi**[¶]**, Stefanie Jegelka**[†]

[†]Massachusetts Institute of Technology (MIT)
[‡]University of Maryland
[§]University of Washington
[¶]National Institute of Informatics (NII)
`keyulu@mit.edu, keyulux@csail.mit.edu`

## Abstract

We study how neural networks trained by gradient descent *extrapolate*, i.e., what they learn outside the support of the training distribution. Previous works report mixed empirical results when extrapolating with neural networks: while feedforward neural networks, a.k.a. multilayer perceptrons (MLPs), do not extrapolate well in certain simple tasks, Graph Neural Networks (GNNs) – structured networks with MLP modules – have shown some success in more complex tasks. Working towards a theoretical explanation, we identify conditions under which MLPs and GNNs extrapolate well. First, we quantify the observation that ReLU MLPs quickly converge to linear functions along any direction from the origin, which implies that ReLU MLPs do not extrapolate most nonlinear functions. But, they can provably learn a linear target function when the training distribution is sufficiently "diverse". Second, in connection to analyzing the successes and limitations of GNNs, these results suggest a hypothesis for which we provide theoretical and empirical evidence: the success of GNNs in extrapolating algorithmic tasks to new data (e.g., larger graphs or edge weights) relies on encoding task-specific non-linearities in the architecture or features. Our theoretical analysis builds on a connection of over-parameterized networks to the neural tangent kernel. Empirically, our theory holds across different training settings.

## 1 Introduction

Humans extrapolate well in many tasks. For example, we can apply arithmetics to arbitrarily large numbers. One may wonder whether a neural network can do the same and generalize to examples arbitrarily far from the training data (Lake et al., 2017). Curiously, previous works report mixed extrapolation results with neural networks. Early works demonstrate feedforward neural networks, a.k.a. multilayer perceptrons (MLPs), fail to extrapolate well when learning simple polynomial functions (Barnard & Wessels, 1992; Haley & Soloway, 1992). However, recent works show Graph Neural Networks (GNNs) (Scarselli et al., 2009), a class of structured networks with MLP building blocks, can generalize to graphs much larger than training graphs in challenging algorithmic tasks, such as predicting the time evolution of physical systems (Battaglia et al., 2016), learning graph algorithms (Velickovic et al., 2020), and solving mathematical equations (Lample & Charton, 2020).

To explain this puzzle, we formally study how neural networks trained by gradient descent (GD) extrapolate, i.e., what they learn outside the support of training distribution. We say a neural network extrapolates well if it learns a task outside the training distribution. At first glance, it may seem that neural networks can behave arbitrarily outside the training distribution since they have high capacity (Zhang et al., 2017) and are universal approximators (Cybenko, 1989; Funahashi, 1989; Hornik et al., 1989; Kurkova, 1992). However, neural networks are constrained by gradient descent training (Hardt et al., 2016; Soudry et al., 2018). In our analysis, we explicitly consider such implicit bias through the analogy of the training dynamics of over-parameterized neural networks and kernel regression via the *neural tangent kernel (NTK)* (Jacot et al., 2018).

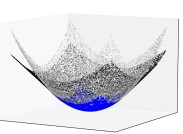 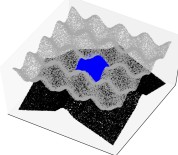 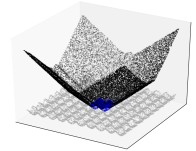 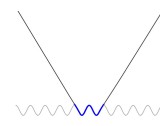

Figure 1: **How ReLU MLPs extrapolate.** We train MLPs to learn nonlinear functions (grey) and plot their predictions both within (blue) and outside (black) the training distribution. MLPs converge quickly to linear functions outside the training data range along directions from the origin (Theorem 1). Hence, MLPs do not extrapolate well in most nonlinear tasks. But, with appropriate training data, MLPs can provably extrapolate linear target functions (Theorem 2).

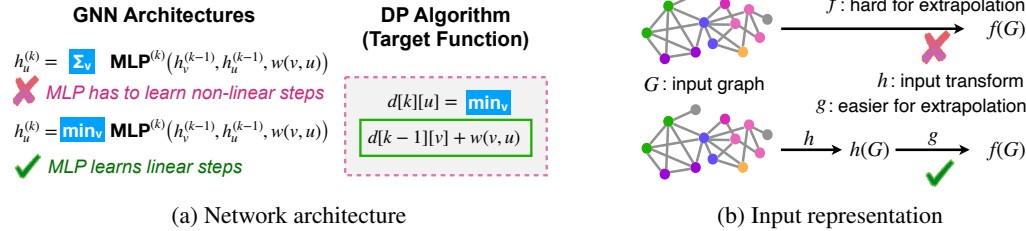

(a) Network architecture                (b) Input representation

Figure 2: **How GNNs extrapolate.** Since MLPs can extrapolate well when learning linear functions, we hypothesize that GNNs can extrapolate well in dynamic programming (DP) tasks if we encode appropriate non-linearities in the architecture (left) and input representation (right; through domain knowledge or representation learning). The encoded non-linearities may not be necessary for interpolation, as they can be approximated by MLP modules, but they help extrapolation. We support the hypothesis theoretically (Theorem 3) and empirically (Figure 6).

Starting with feedforward networks, the simplest neural networks and building blocks of more complex architectures such as GNNs, we establish that the predictions of over-parameterized MLPs with ReLU activation trained by GD converge to linear functions along any direction from the origin. We prove a convergence rate for two-layer networks and empirically observe that convergence often occurs *close to* the training data (Figure 1), which suggests ReLU MLPs cannot extrapolate well for most nonlinear tasks. We emphasize that our results do not follow from the fact that ReLU networks have finitely many linear regions (Arora et al., 2018; Hanin & Rolnick, 2019; Hein et al., 2019). While having finitely many linear regions implies ReLU MLPs *eventually* become linear, it does not say whether MLPs will learn the correct target function close to the training distribution. In contrast, our results are non-asymptotic and quantify what kind of functions MLPs will learn close to the training distribution. Second, we identify a condition when MLPs extrapolate well: the task is linear and the geometry of the training distribution is sufficiently "diverse". To our knowledge, our results are the first extrapolation results of this kind for feedforward neural networks.

We then relate our insights into feedforward neural networks to GNNs, to explain why GNNs extrapolate well in some algorithmic tasks. Prior works report successful extrapolation for tasks that can be solved by dynamic programming (DP) (Bellman, 1966), which has a computation structure aligned with GNNs (Xu et al., 2020). DP updates can often be decomposed into nonlinear and linear steps. Hence, we hypothesize that GNNs trained by GD can extrapolate well in a DP task, if we encode appropriate non-linearities in the *architecture* and *input representation* (Figure 2). Importantly, encoding non-linearities may be unnecessary for GNNs to *interpolate*, because the MLP modules can easily learn many nonlinear functions inside the training distribution (Cybenko, 1989; Hornik et al., 1989; Xu et al., 2020), but it is crucial for GNNs to *extrapolate* correctly. We prove this hypothesis for a simplified case using *Graph NTK* (Du et al., 2019b). Empirically, we validate the hypothesis on three DP tasks: max degree, shortest paths, and $n$-body problem. We show GNNs with appropriate architecture, input representation, and training distribution can predict well on graphs with unseen sizes, structures, edge weights, and node features. Our theory explains the empirical success in previous works and suggests their limitations: successful extrapolation relies on encoding task-specific non-linearities, which requires domain knowledge or extensive model search. From a broader standpoint, our insights go beyond GNNs and apply broadly to other neural networks.

To summarize, we study how neural networks extrapolate. First, ReLU MLPs trained by GD converge to linear functions along directions from the origin with a rate of $O(1/t)$. Second, to explain why GNNs extrapolate well in some algorithmic tasks, we prove that ReLU MLPs can extrapolate well in linear tasks, leading to a hypothesis: a neural network can extrapolate well when appropriate non-linearities are encoded into the architecture and features. We prove this hypothesis for a simplified case and provide empirical support for more general settings.

## 1.1 RELATED WORK

Early works show example tasks where MLPs do not extrapolate well, e.g. learning simple polynomials (Barnard & Wessels, 1992; Haley & Soloway, 1992). We instead show a *general* pattern of how ReLU MLPs extrapolate and identify conditions for MLPs to extrapolate well. More recent works study the implicit biases induced on MLPs by gradient descent, for both the NTK and mean field regimes (Bietti & Mairal, 2019; Chizat & Bach, 2018; Song et al., 2018). Related to our results, some works show MLP predictions converge to "simple" piecewise linear functions, e.g., with few linear regions (Hanin & Rolnick, 2019; Maennel et al., 2018; Savarese et al., 2019; Williams et al., 2019). Our work differs in that none of these works explicitly studies extrapolation, and some focus only on one-dimensional inputs. Recent works also show that in high-dimensional settings of the NTK regime, MLP is asymptotically at most a linear predictor in certain scaling limits (Ba et al., 2020; Ghorbani et al., 2019). We study a different setting (extrapolation), and our analysis is non-asymptotic in nature and does not rely on random matrix theory.

Prior works explore GNN extrapolation by testing on larger graphs (Battaglia et al., 2018; Santoro et al., 2018; Saxton et al., 2019; Velickovic et al., 2020). We are the first to theoretically study GNN extrapolation, and we complete the notion of extrapolation to include unseen features and structures.

## 2 PRELIMINARIES

We begin by introducing our setting. Let $\mathcal{X}$ be the domain of interest, e.g., vectors or graphs. The task is to learn an underlying function $g : \mathcal{X} \to \mathbb{R}$ with a training set $\{(\boldsymbol{x}_i, y_i)\}_{i=1}^n \subset \mathcal{D}$, where $y_i = g(\boldsymbol{x}_i)$ and $\mathcal{D}$ is the support of training distribution. Previous works have extensively studied in-distribution generalization where the training and the test distributions are identical (Valiant, 1984; Vapnik, 2013); i.e., $\mathcal{D} = \mathcal{X}$. In contrast, extrapolation addresses predictions on a domain $\mathcal{X}$ that is larger than the support of the training distribution $\mathcal{D}$. We will say a model *extrapolates well* if it has a small *extrapolation error*.

**Definition 1.** (Extrapolation error). Let $f : \mathcal{X} \to \mathbb{R}$ be a model trained on $\{(\boldsymbol{x}_i, y_i)\}_{i=1}^n \subset \mathcal{D}$ with underlying function $g : \mathcal{X} \to \mathbb{R}$. Let $\mathcal{P}$ be a distribution over $\mathcal{X} \setminus \mathcal{D}$ and let $\ell : \mathbb{R} \times \mathbb{R} \to \mathbb{R}$ be a loss function. We define the extrapolation error of $f$ as $\mathbb{E}_{\boldsymbol{x} \sim \mathcal{P}}[\ell(f(\boldsymbol{x}), g(\boldsymbol{x}))]$.

We focus on neural networks trained by *gradient descent* (GD) or its variants with *squared loss*. We study two network architectures: feedforward and graph neural networks.

**Graph Neural Networks.** GNNs are structured networks operating on graphs with MLP modules (Battaglia et al., 2018; Xu et al., 2019). Let $G = (V, E)$ be a graph. Each node $u \in V$ has a feature vector $\boldsymbol{x}_u$, and each edge $(u, v) \in E$ has a feature vector $\boldsymbol{w}_{(u,v)}$. GNNs recursively compute node representations $\boldsymbol{h}_u^{(k)}$ at iteration $k$ (Gilmer et al., 2017; Xu et al., 2018). Initially, $\boldsymbol{h}_u^{(0)} = \boldsymbol{x}_u$. For $k = 1..K$, GNNs update $\boldsymbol{h}_u^{(k)}$ by aggregating the neighbor representations. We can optionally compute a graph representation $\boldsymbol{h}_G$ by aggregating the final node representations. That is,

$$\boldsymbol{h}_u^{(k)} = \sum_{v \in \mathcal{N}(u)} \text{MLP}^{(k)}\Big(\boldsymbol{h}_u^{(k-1)}, \boldsymbol{h}_v^{(k-1)}, \boldsymbol{w}_{(v,u)}\Big), \quad \boldsymbol{h}_G = \text{MLP}^{(K+1)}\Big(\sum_{u \in G} \boldsymbol{h}_u^{(K)}\Big). \quad (1)$$

The final output is the graph representation $\boldsymbol{h}_G$ or final node representations $\boldsymbol{h}_u^{(K)}$ depending on the task. We refer to the neighbor aggregation step for $\boldsymbol{h}_u^{(k)}$ as *aggregation* and the pooling step in $\boldsymbol{h}_G$ as *readout*. Previous works typically use sum-aggregation and sum-readout (Battaglia et al., 2018). Our results indicate why replacing them may help extrapolation (Section 4).

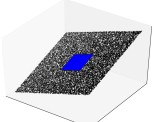 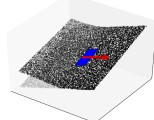 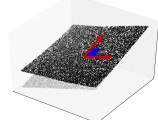 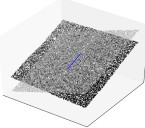

Figure 3: **Conditions for ReLU MLPs to extrapolate well.** We train MLPs to learn linear functions (grey) with different training distributions (blue) and plot out-of-distribution predictions (black). Following Theorem 2, MLPs extrapolate well when the training distribution (blue) has support in all directions (first panel), but not otherwise: in the two middle panels, some dimensions of the training data are constrained to be positive (red arrows); in the last panel, one dimension is a fixed constant.

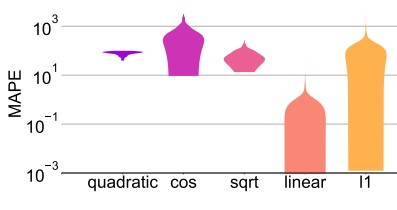

(a) Different target functions

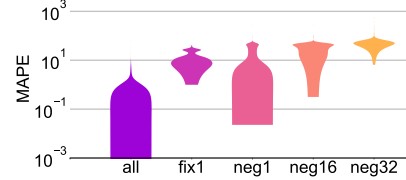

(b) Different training distributions for linear target

Figure 4: **Extrapolation performance of ReLU MLPs.** We plot the distributions of MAPE (mean absolute percentage error) of MLPs trained with various hyperparameters (depth, width, learning rate, batch size). (a) Learning different target functions; (b) Different training distributions for learning linear target functions: "all" covers all directions, "fix1" has one dimension fixed to a constant, and "neg$d$" has $d$ dimensions constrained to negative values. ReLU MLPs generally do not extrapolate well unless the target function is linear along each direction (Figure 4a), and extrapolate linear target functions if the training distribution covers sufficiently many directions (Figure 4b).

## 3 How Feedforward Neural Networks Extrapolate

Feedforward networks are the simplest neural networks and building blocks of more complex architectures such as GNNs, so we first study how they extrapolate when trained by GD. Throughout the paper, we assume ReLU activation. Section 3.3 contains preliminary results for other activations.

### 3.1 Linear Extrapolation Behavior of ReLU MLPs

By architecture, ReLU networks learn piecewise linear functions, but what do these regions precisely look like outside the support of the training data? Figure 1 illustrates examples of how ReLU MLPs extrapolate when trained by GD on various nonlinear functions. These examples suggest that outside the training support, the predictions quickly become linear along directions from the origin. We systematically verify this pattern by linear regression on MLPs' predictions: the coefficient of determination ($R^2$) is always greater than 0.99 (Appendix C.2). That is, ReLU MLPs "linearize" almost immediately outside the training data range.

We formalize this observation using the implicit biases of neural networks trained by GD via the *neural tangent kernel (NTK)*: optimization trajectories of over-parameterized networks trained by GD are equivalent to those of kernel regression with a specific neural tangent kernel, under a set of assumptions called the "NTK regime" (Jacot et al., 2018). We provide an informal definition here; for further details, we refer the readers to Jacot et al. (2018) and Appendix A.

**Definition 2.** (Informal) A neural network trained in the *NTK regime* is infinitely wide, randomly initialized with certain scaling, and trained by GD with infinitesimal steps.

Prior works analyze optimization and in-distribution generalization of over-parameterized neural networks via NTK (Allen-Zhu et al., 2019a;b; Arora et al., 2019a;b; Cao & Gu, 2019; Du et al., 2019c;a; Li & Liang, 2018; Nitanda & Suzuki, 2021). We instead analyze extrapolation.

Theorem 1 formalizes our observation from Figure 1: outside the training data range, along any direction $t\boldsymbol{v}$ from the origin, the prediction of a two-layer ReLU MLP quickly converges to a linear

function with rate $O(\frac{1}{t})$. The linear coefficients $\boldsymbol{\beta_v}$ and the constant terms in the convergence rate depend on the training data and direction $\boldsymbol{v}$. The proof is in Appendix B.1.

**Theorem 1.** *(Linear extrapolation). Suppose we train a two-layer ReLU MLP $f : \mathbb{R}^d \to \mathbb{R}$ with squared loss in the NTK regime. For any direction $\boldsymbol{v} \in \mathbb{R}^d$, let $\boldsymbol{x}_0 = t\boldsymbol{v}$. As $t \to \infty$, $f(\boldsymbol{x}_0 + h\boldsymbol{v}) - f(\boldsymbol{x}_0) \to \beta_{\boldsymbol{v}} \cdot h$ for any $h > 0$, where $\beta_{\boldsymbol{v}}$ is a constant linear coefficient. Moreover, given $\epsilon > 0$, for $t = O(\frac{1}{\epsilon})$, we have $|\frac{f(\boldsymbol{x}_0 + h\boldsymbol{v}) - f(\boldsymbol{x}_0)}{h} - \beta_{\boldsymbol{v}}| < \epsilon$.*

ReLU networks have finitely many linear regions (Arora et al., 2018; Hanin & Rolnick, 2019), hence their predictions *eventually* become linear. In contrast, Theorem 1 is a more fine-grained analysis of *how* MLPs extrapolate and provides a convergence rate. While Theorem 1 assumes two-layer networks in the NTK regime, experiments confirm that the linear extrapolation behavior happens across networks with different depths, widths, learning rates, and batch sizes (Appendix C.1 and C.2). Our proof technique potentially also extends to deeper networks.

Theorem 1 implies which target functions a ReLU MLP may be able to match outside the training data: only functions that are almost-linear along the directions away from the origin. Indeed, Figure 4a shows ReLU MLPs do not extrapolate target functions such as $\boldsymbol{x}^\top A\boldsymbol{x}$ (quadratic), $\sum_{i=1}^d \cos(2\pi \cdot \boldsymbol{x}^{(i)})$ (cos), and $\sum_{i=1}^d \sqrt{\boldsymbol{x}^{(i)}}$ (sqrt), where $\boldsymbol{x}^{(i)}$ is the $i$-th dimension of $\boldsymbol{x}$. With suitable hyperparameters, MLPs extrapolate the L1 norm correctly, which satisfies the directional linearity condition.

Figure 4a provides one more positive result: MLPs extrapolate linear target functions well, across many different hyperparameters. While learning linear functions may seem very limited at first, in Section 4 this insight will help explain extrapolation properties of GNNs in non-linear practical tasks. Before that, we first theoretically analyze when MLPs extrapolate well.

## 3.2 When ReLU MLPs Provably Extrapolate Well

Figure 4a shows that MLPs can extrapolate well when the target function is linear. However, this is not always true. In this section, we show that successful extrapolation depends on the *geometry* of training data. Intuitively, the training distribution must be "diverse" enough for correct extrapolation.

We provide two conditions that relate the geometry of the training data to extrapolation. Lemma 1 states that over-parameterized MLPs can learn a linear target function with only $2d$ examples.

**Lemma 1.** *Let $g(\boldsymbol{x}) = \boldsymbol{\beta}^\top \boldsymbol{x}$ be the target function for $\boldsymbol{\beta} \in \mathbb{R}^d$. Suppose $\{\boldsymbol{x}_i\}_{i=1}^n$ contains an orthogonal basis $\{\hat{\boldsymbol{x}}_i\}_{i=1}^d$ and $\{-\hat{\boldsymbol{x}}_i\}_{i=1}^d$. If we train a two-layer ReLU MLP $f$ on $\{(\boldsymbol{x}_i, y_i)\}_{i=1}^n$ with squared loss in the NTK regime, then $f(\boldsymbol{x}) = \boldsymbol{\beta}^\top \boldsymbol{x}$ for all $\boldsymbol{x} \in \mathbb{R}^d$.*

Lemma 1 is mainly of theoretical interest, as the $2d$ examples need to be carefully chosen. Theorem 2 builds on Lemma 1 and identifies a more practical condition for successful extrapolation: if the support of the training distribution covers all directions (e.g., a hypercube that covers the origin), the MLP converges to a linear target function with sufficient training data.

**Theorem 2.** *(Conditions for extrapolation). Let $g(\boldsymbol{x}) = \boldsymbol{\beta}^\top \boldsymbol{x}$ be the target function for $\boldsymbol{\beta} \in \mathbb{R}^d$. Suppose $\{\boldsymbol{x}_i\}_{i=1}^n$ is sampled from a distribution whose support $\mathcal{D}$ contains a connected subset $\mathcal{S}$, where for any non-zero $\boldsymbol{w} \in \mathbb{R}^d$, there exists $k > 0$ so that $k\boldsymbol{w} \in \mathcal{S}$. If we train a two-layer ReLU MLP $f : \mathbb{R}^d \to \mathbb{R}$ on $\{(\boldsymbol{x}_i, y_i)\}_{i=1}^n$ with squared loss in the NTK regime, $f(\boldsymbol{x}) \xrightarrow{p} \boldsymbol{\beta}^\top \boldsymbol{x}$ as $n \to \infty$.*

**Experiments: geometry of training data affects extrapolation.** The condition in Theorem 2 formalizes the intuition that the training distribution must be "diverse" for successful extrapolation, e.g., $\mathcal{D}$ includes all directions. Empirically, the extrapolation error is indeed small when the condition of Theorem 2 is satisfied ("all" in Figure 4b). In contrast, the extrapolation error is much larger when the training examples are restricted to only some directions (Figure 4b and Figure 3).

Relating to previous works, Theorem 2 suggests why spurious correlations may hurt extrapolation, complementing the causality arguments (Arjovsky et al., 2019; Peters et al., 2016; Rojas-Carulla et al., 2018). When the training data has spurious correlations, some combinations of features are missing; e.g., camels might only appear in deserts in an image collection. Therefore, the condition for Theorem 2 no longer holds, and the model may extrapolate incorrectly. Theorem 2 is also analogous to an identifiability condition for linear models, but stricter. We can uniquely identify a linear function if the training data has full (feature) rank. MLPs are more expressive, so identifying the linear target function requires additional constraints.

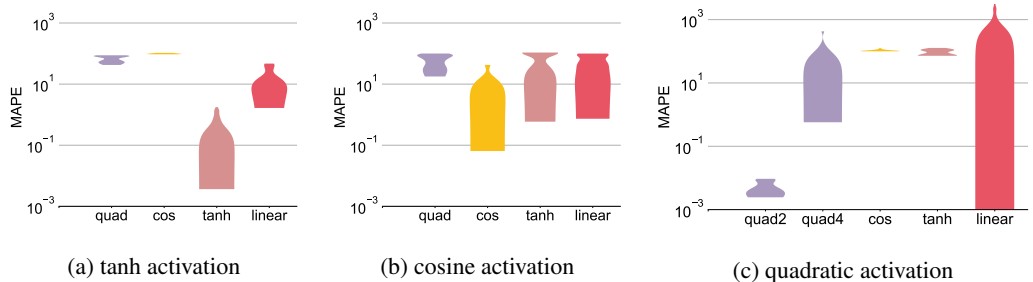

(a) tanh activation     (b) cosine activation     (c) quadratic activation

Figure 5: **Extrapolation performance of MLPs with other activation**. MLPs can extrapolate well when the activation is "similar" to the target function. When learning quadratic with quadratic activation, 2-layer networks (quad-2) extrapolate well, but 4-layer networks (quad-4) do not.

To summarize, we analyze how ReLU MLPs extrapolate and provide two insights: (1) MLPs cannot extrapolate most nonlinear tasks due to their linear extrapolation (Theorem 1); and (2) MLPs extrapolate well when the target function is linear, if the training distribution is "diverse" (Theorem 2). In the next section, these results help us understand how more complex networks extrapolate.

### 3.3 MLPs with Other Activation Functions

Before moving on to GNNs, we complete the picture of MLPs with experiments on other activation functions: tanh $\sigma(x) = \tanh(x)$, cosine $\sigma(x) = \cos(x)$ (Lapedes & Farber, 1987; McCaughan, 1997; Sopena & Alquezar, 1994), and quadratic $\sigma(x) = x^2$ (Du & Lee, 2018; Livni et al., 2014). Details are in Appendix C.4. MLPs extrapolate well when the activation and target function are similar; e.g., tanh activation extrapolates well when learning tanh, but not other functions (Figure 5). Moreover, each activation function has different limitations. To extrapolate the tanh function with tanh activation, the training data range has to be sufficiently wide. When learning a quadratic function with quadratic activation, only two-layer networks extrapolate well as more layers lead to higher-order polynomials. Cosine activations are hard to optimize for high-dimensional data, so we only consider one/two dimensional cosine target functions.

## 4 How Graph Neural Networks Extrapolate

Above, we saw that extrapolation in nonlinear tasks is hard for MLPs. Despite this limitation, GNNs have been shown to extrapolate well in some nonlinear algorithmic tasks, such as intuitive physics (Battaglia et al., 2016; Janner et al., 2019), graph algorithms (Battaglia et al., 2018; Velickovic et al., 2020), and symbolic mathematics (Lample & Charton, 2020). To address this discrepancy, we build on our MLP results and study how GNNs trained by GD extrapolate.

### 4.1 Hypothesis: Linear Algorithmic Alignment Helps Extrapolation

We start with an example: training GNNs to solve the shortest path problem. For this task, prior works observe that a modified GNN architecture with min-aggregation can generalize to graphs larger than those in the training set (Battaglia et al., 2018; Velickovic et al., 2020):

$$\boldsymbol{h}_u^{(k)} = \min_{v \in \mathcal{N}(u)} \text{MLP}^{(k)}\big(\boldsymbol{h}_u^{(k-1)}, \boldsymbol{h}_v^{(k-1)}, \boldsymbol{w}_{(v,u)}\big). \tag{2}$$

We first provide an intuitive explanation (Figure 2a). Shortest path can be solved by the Bellman-Ford (BF) algorithm (Bellman, 1958) with the following update:

$$d[k][u] = \min_{v \in \mathcal{N}(u)} d[k-1][v] + \boldsymbol{w}(v,u), \tag{3}$$

where $\boldsymbol{w}(v,u)$ is the weight of edge $(v,u)$, and $d[k][u]$ is the shortest distance to node $u$ within $k$ steps. The two equations can be easily aligned: GNNs simulate the BF algorithm if its MLP modules learn a linear function $d[k-1][v]+\boldsymbol{w}(v,u)$. Since MLPs can extrapolate linear tasks, this "alignment" may explain why min-aggregation GNNs can extrapolate well in this task.

For comparison, we can reason why we would not expect GNNs with the more commonly used sum-aggregation (Eqn. 1) to extrapolate well in this task. With sum-aggregation, the MLP modules need to learn a nonlinear function to simulate the BF algorithm, but Theorem 1 suggests that they will not extrapolate most nonlinear functions outside the training support.

We can generalize the above intuition to other algorithmic tasks. Many tasks where GNNs extrapolate well can be solved by dynamic programming (DP) (Bellman, 1966), an algorithmic paradigm with a recursive structure similar to GNNs' (Eqn. 1) (Xu et al., 2020).

**Definition 3.** *Dynamic programming* (DP) is a recursive procedure with updates

$$\text{Answer}[k][s] = \text{DP-Update}(\{\text{Answer}[k-1][s']\}, s' = 1...n), \tag{4}$$

where $\text{Answer}[k][s]$ is the solution to a sub-problem indexed by iteration $k$ and state $s$, and DP-Update is a task-specific update function that solves the sub-problem based on the previous iteration.

From a broader standpoint, we hypothesize that: if we encode appropriate non-linearities into the model architecture and input representations so that the MLP modules only need to learn nearly linear steps, then the resulting neural network can extrapolate well.

**Hypothesis 1.** (Linear algorithmic alignment). Let $f : \mathcal{X} \to \mathbb{R}$ be the underlying function and $\mathcal{N}$ a neural network with $m$ MLP modules. Suppose there exist $m$ *linear* functions $\{g_i\}_{i=1}^m$ so that by replacing $\mathcal{N}$'s MLP modules with $g_i$'s, $\mathcal{N}$ simulates $f$. Given $\epsilon > 0$, there exists $\{(x_i, f(x_i))\}_{i=1}^n \subset \mathcal{D} \subsetneq \mathcal{X}$ so that $\mathcal{N}$ trained on $\{(x_i, f(x_i))\}_{i=1}^n$ by GD with squared loss learns $\hat{f}$ with $\|\hat{f} - f\| < \epsilon$.

Our hypothesis builds on the algorithmic alignment framework of (Xu et al., 2020), which states that a neural network *interpolates* well if the modules are "aligned" to easy-to-learn (possibly nonlinear) functions. Successful extrapolation is harder: the modules need to align with linear functions.

**Applications of linear algorithmic alignment.** In general, linear algorithmic alignment is not restricted to GNNs and applies broadly to neural networks. To satisfy the condition, we can encode appropriate nonlinear operations in the *architecture* or *input representation* (Figure 2). Learning DP algorithms with GNNs is one example of encoding non-linearity in the architecture (Battaglia et al., 2018; Corso et al., 2020). Another example is to encode log-and-exp transforms in the architecture to help extrapolate multiplication in arithmetic tasks (Trask et al., 2018; Madsen & Johansen, 2020). Neural symbolic programs take a step further and encode a library of symbolic operations to help extrapolation (Johnson et al., 2017; Mao et al., 2019; Yi et al., 2018).

For some tasks, it may be easier to change the input representation (Figure 2b). Sometimes, we can decompose the target function $f$ as $f = g \circ h$ into a feature embedding $h$ and a "simpler" target function $g$ that our model can extrapolate well. We can obtain $h$ via specialized features or feature transforms using *domain knowledge* (Lample & Charton, 2020; Webb et al., 2020), or via *representation learning* (e.g., BERT) with unlabeled out-of-distribution data in $\mathcal{X} \setminus \mathcal{D}$ (Chen et al., 2020; Devlin et al., 2019; Hu et al., 2020; Mikolov et al., 2013b; Peters et al., 2018). This brings a new perspective of how representations help extrapolation in various application areas. For example, in natural language processing, pretrained representations (Mikolov et al., 2013a; Wu & Dredze, 2019) and feature transformation using domain knowledge (Yuan et al., 2020; Zhang et al., 2019) help models generalize across languages, a special type of extrapolation. In quantitative finance, identifying the right "factors" or features is crucial for deep learning models as the financial markets may frequently be in extrapolation regimes (Banz, 1981; Fama & French, 1993; Ross, 1976).

Linear algorithmic alignment explains successful extrapolation in the literature and suggests that extrapolation is harder in general: encoding appropriate non-linearity often requires domain expertise or model search. Next, we provide theoretical and empirical support for our hypothesis.

## 4.2 THEORETICAL AND EMPIRICAL SUPPORT

We validate our hypothesis on three DP tasks: max degree, shortest path, and $n$-body problem, and prove the hypothesis for max degree. We highlight the role of graph structures in extrapolation.

**Theoretical analysis.** We start with a simple yet fundamental task: learning the max degree of a graph, a special case of DP with one iteration. As a corollary of Theorem 1, the commonly used sum-based GNN (Eqn. 1) cannot extrapolate well (proof in Appendix B.4).

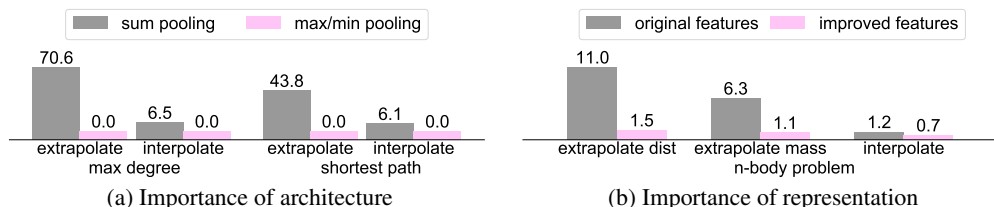

(a) Importance of architecture  (b) Importance of representation

Figure 6: **Extrapolation for algorithmic tasks.** Each column indicates the task and mean average percentage error (MAPE). Encoding appropriate non-linearity in the architecture or representation is less helpful for *interpolation*, but significantly improves *extrapolation*. Left: In max degree and shortest path, GNNs that appropriately encode max/min extrapolate well, but GNNs with sum-pooling do not. Right: With improved input representation, GNNs extrapolate better for the $n$-body problem.

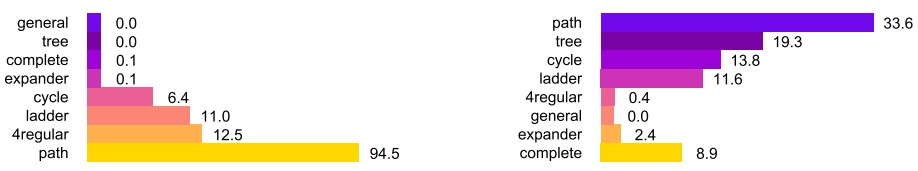

(a) Max degree with GNNs that encode max  (b) Shortest path with GNNs that encode min

Figure 7: **Importance of the training graph structure.** Rows indicate the graph structure covered by the training set and the extrapolation error (MAPE). In max degree, GNNs with max readout extrapolate well if the max/min degrees of the training graphs are not restricted (Theorem 3). In shortest path, the extrapolation errors of min GNNs follow a U-shape in the sparsity of the training graphs. More results may be found in Appendix D.2.

**Corollary 1.** *GNNs with sum-aggregation and sum-readout do not extrapolate well in Max Degree.*

To achieve linear algorithmic alignment, we can encode the only non-linearity, the max function, in the readout. Theorem 3 confirms that a GNN with max-readout can extrapolate well in this task.

**Theorem 3.** *(Extrapolation with GNNs). Assume all nodes have the same feature. Let $g$ and $g'$ be the max/min degree function, respectively. Let $\{(G_i, g(G_i)\}_{i=1}^n$ be the training set. If $\{(g(G_i), g'(G_i), g(G_i) \cdot N_i^{\max}, g'(G_i) \cdot N_i^{\min})\}_{i=1}^n$ spans $\mathbb{R}^4$, where $N_i^{\max}$ and $N_i^{\min}$ are the number of nodes that have max/min degree on $G_i$, then one-layer max-readout GNNs trained on $\{(G_i, g(G_i))\}_{i=1}^n$ with squared loss in the NTK regime learn $g$.*

Theorem 3 does not follow immediately from Theorem 2, because MLP modules in GNNs only receive indirect supervision. We analyze the *Graph NTK* (Du et al., 2019b) to prove Theorem 3 in Appendix B.5. While Theorem 3 assumes identical node features, we empirically observe similar results for both identical and non-identical features (Figure 16 in Appendix).

**Interpretation of conditions.** The condition in Theorem 3 is analogous to that in Theorem 2. Both theorems require diverse training data, measured by *graph structure* in Theorem 3 or *directions* in Theorem 2. In Theorem 3, the condition is violated if all training graphs have the same max or min node degrees, e.g., when training data are from one of the following families: path, $C$-regular graphs (regular graphs with degree $C$), cycle, and ladder.

**Experiments: architectures that help extrapolation.** We validate our theoretical analysis with two DP tasks: max degree and shortest path (details in Appendix C.5 and C.6). While previous works only test on graphs with different sizes (Battaglia et al., 2018; Velickovic et al., 2020), we also test on graphs with unseen structure, edge weights and node features. The results support our theory. For max degree, GNNs with max-readout are better than GNNs with sum-readout (Figure 6a), confirming Corollary 1 and Theorem 3. For shortest path, GNNs with min-readout and min-aggregation are better than GNNs with sum-readout (Figure 6a).

Experiments confirm the importance of training graphs structure (Figure 7). Interestingly, the two tasks favor different graph structure. For max degree, as Theorem 3 predicts, GNNs extrapolate well when trained on trees, complete graphs, expanders, and general graphs, and extrapolation errors are

higher when trained on 4-regular, cycles, or ladder graphs. For shortest path, extrapolation errors follow a U-shaped curve as we change the *sparsity* of training graphs (Figure 7b and Figure 18 in Appendix). Intuitively, models trained on sparse or dense graphs likely learn degenerative solutions.

**Experiments: representations that help extrapolation.** Finally, we show a good input representation helps extrapolation. We study the $n$-body problem (Battaglia et al., 2016; Watters et al., 2017) (Appendix C.7), that is, predicting the time evolution of $n$ objects in a gravitational system. Following previous work, the input is a complete graph where the nodes are the objects (Battaglia et al., 2016). The node feature for $u$ is the concatenation of the object's mass $m_u$, position $\boldsymbol{x}_u^{(t)}$, and velocity $\boldsymbol{v}_u^{(t)}$ at time $t$. The edge features are set to zero. We train GNNs to predict the velocity of each object $u$ at time $t+1$. The true velocity $f(G; u)$ for object $u$ is approximately

$$f(G; u) \approx \boldsymbol{v}_u^t + \boldsymbol{a}_u^t \cdot dt, \quad \boldsymbol{a}_u^t = C \cdot \sum_{v \neq u} \frac{m_v}{\|\boldsymbol{x}_u^t - \boldsymbol{x}_v^t\|_2^3} \cdot \left(\boldsymbol{x}_v^t - \boldsymbol{x}_u^t\right), \tag{5}$$

where $C$ is a constant. To learn $f$, the MLP modules need to learn a nonlinear function. Therefore, GNNs do not extrapolate well to unseen masses or distances ("original features" in Figure 6b). We instead use an improved representation $h(G)$ to encode non-linearity. At time $t$, we transform the edge features of $(u, v)$ from zero to $\boldsymbol{w}_{(u,v)}^{(t)} = m_v \cdot \left(\boldsymbol{x}_v^{(t)} - \boldsymbol{x}_u^{(t)}\right) / \|\boldsymbol{x}_u^{(t)} - \boldsymbol{x}_v^{(t)}\|_2^3$. The new edge features do not add information, but the MLP modules now only need to learn linear functions, which helps extrapolation ("improved features" in Figure 6b).

## 5    CONNECTIONS TO OTHER OUT-OF-DISTRIBUTION SETTINGS

We discuss several related settings. Intuitively, from the viewpoint of our results above, methods in related settings may improve extrapolation by 1) learning useful non-linearities beyond the training data range and 2) mapping relevant test data to the training data range.

**Domain adaptation** studies generalization to a specific target domain (Ben-David et al., 2010; Blitzer et al., 2008; Mansour et al., 2009). Typical strategies adjust the training process: for instance, use unlabeled samples from the target domain to align the target and source distributions (Ganin et al., 2016; Zhao et al., 2018). Using target domain data during training may induce useful non-linearities and may mitigate extrapolation by matching the target and source distributions, though the correctness of the learned mapping depends on the label distribution (Zhao et al., 2019).

**Self-supervised learning** on a large amount of unlabeled data can learn useful non-linearities beyond the labeled training data range (Chen et al., 2020; Devlin et al., 2019; He et al., 2020; Peters et al., 2018). Hence, our results suggest an explanation why pre-trained representations such as BERT improve out-of-distribution robustness (Hendrycks et al., 2020). In addition, self-supervised learning could map semantically similar data to similar representations, so some out-of-domain examples might fall inside the training distribution after the mapping.

**Invariant models** aim to learn features that respect specific invariances across multiple training distributions (Arjovsky et al., 2019; Rojas-Carulla et al., 2018; Zhou et al., 2021). If the model indeed learns these invariances, which can happen in the linear case and when there are confounders or anti-causal variables (Ahuja et al., 2021; Rosenfeld et al., 2021), this may essentially increase the training data range, since variations in the invariant features may be ignored by the model.

**Distributional robustness** considers small adversarial perturbations of the data distribution, and ensures that the model performs well under these (Goh & Sim, 2010; Sagawa et al., 2020; Sinha et al., 2018; Staib & Jegelka, 2019). We instead look at more global perturbations. Still, one would expect that modifications that help extrapolation in general also improve robustness to local perturbations.

## 6    CONCLUSION

This paper is an initial step towards formally understanding how neural networks trained by gradient descent extrapolate. We identify conditions under which MLPs and GNNs extrapolate as desired. We also suggest an explanation how GNNs have been able to extrapolate well in complex algorithmic tasks: encoding appropriate non-linearity in architecture and features can help extrapolation. Our results and hypothesis agree with empirical results, in this paper and in the literature.

ACKNOWLEDGMENTS

We thank Ruosong Wang, Tianle Cai, Han Zhao, Yuichi Yoshida, Takuya Konishi, Toru Lin, Weihua Hu, Matt J. Staib, Yichao Zhou, Denny Wu, Tianyi Yang, and Dingli (Leo) Yu for insightful discussions. This research was supported by NSF CAREER award 1553284, NSF III 1900933, and a Chevron-MIT Energy Fellowship. This research was also supported by JST ERATO JPMJER1201 and JSPS Kakenhi JP18H05291. MZ was supported by ODNI, IARPA, via the BETTER Program contract 2019-19051600005. The views, opinions, and/or findings contained in this article are those of the author and should not be interpreted as representing the official views or policies, either expressed or implied, of the Defense Advanced Research Projects Agency, the Department of Defense, ODNI, IARPA, or the U.S. Government. The U.S. Government is authorized to reproduce and distribute reprints for governmental purposes notwithstanding any copyright annotation therein.

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

## A    THEORETICAL BACKGROUND

In this section, we introduce theoretical background on neural tangent kernel (NTK), which draws an equivalence between the training dynamics of infinitely-wide (or ultra-wide) neural networks and that of kernel regression with respect to the neural tangent kernel.

Consider a general neural network $f(\boldsymbol{\theta}, \boldsymbol{x}) : \mathcal{X} \to \mathbb{R}$ where $\boldsymbol{\theta} \in \mathbb{R}^m$ is the parameters in the network and $\boldsymbol{x} \in \mathcal{X}$ is the input. Suppose we train the neural network by minimizing the squared loss over training data, $\ell(\boldsymbol{\theta}) = \frac{1}{2} \sum_{i=1}^{n} (f(\boldsymbol{\theta}, \boldsymbol{x}_i) - y_i)^2$, by gradient descent with infinitesimally small learning rate, i.e., $\frac{d\boldsymbol{\theta}(t)}{dt} = -\nabla \ell(\boldsymbol{\theta}(t))$. Let $\boldsymbol{u}(t) = (f(\boldsymbol{\theta}(t), \boldsymbol{x}_i))_{i=1}^{n}$ be the network outputs. $\boldsymbol{u}(t)$ follows the dynamics

$$\frac{d\boldsymbol{u}(t)}{dt} = -\boldsymbol{H}(t)(\boldsymbol{u}(t) - y), \tag{6}$$

where $\boldsymbol{H}(t)$ is an $n \times n$ matrix whose $(i, j)$-th entry is

$$\boldsymbol{H}(t)_{ij} = \left\langle \frac{\partial f(\boldsymbol{\theta}(t), \boldsymbol{x}_i)}{\partial \boldsymbol{\theta}}, \frac{\partial f(\boldsymbol{\theta}(t), \boldsymbol{x}_j)}{\partial \boldsymbol{\theta}} \right\rangle. \tag{7}$$

A line of works show that for sufficiently wide networks, $\boldsymbol{H}(t)$ stays almost constant during training, i.e., $\boldsymbol{H}(t) = \boldsymbol{H}(0)$ in the limit (Arora et al., 2019a;b; Allen-Zhu et al., 2019a; Du et al., 2019c;a; Li & Liang, 2018; Jacot et al., 2018). Suppose network parameters are randomly initialized with certain scaling, as network width goes to infinity, $\boldsymbol{H}(0)$ converges to a fixed matrix, the neural tangent kernel (NTK) (Jacot et al., 2018):

$$\text{NTK}(\boldsymbol{x}, \boldsymbol{x}') = \mathop{\mathbb{E}}_{\boldsymbol{\theta} \sim \mathcal{W}} \left\langle \frac{\partial f(\boldsymbol{\theta}(t), \boldsymbol{x})}{\partial \boldsymbol{\theta}}, \frac{\partial f(\boldsymbol{\theta}(t), \boldsymbol{x}')}{\partial \boldsymbol{\theta}} \right\rangle, \tag{8}$$

where $\mathcal{W}$ is Gaussian.

Therefore, the learning dynamics of sufficiently wide neural networks in this regime is equivalent to that of kernel gradient descent with respect to the NTK. This implies the function learned by a neural network at convergence on any specific training set, denoted by $f_{\text{NTK}}(\boldsymbol{x})$, can be precisely characterized, and is equivalent to the following kernel regression solution

$$f_{\text{NTK}}(\boldsymbol{x}) = (\text{NTK}(\boldsymbol{x}, \boldsymbol{x}_1), ..., \text{NTK}(\boldsymbol{x}, \boldsymbol{x}_n)) \cdot \text{NTK}_{\text{train}}^{-1} \boldsymbol{Y}, \tag{9}$$

where $\text{NTK}_{\text{train}}$ is the $n \times n$ kernel for training data, $\text{NTK}(\boldsymbol{x}, \boldsymbol{x}_i)$ is the kernel value between test data $\boldsymbol{x}$ and training data $\boldsymbol{x}_i$, and $\boldsymbol{Y}$ is the training labels.

We can in fact exactly calculate the neural tangent kernel matrix for certain architectures and activation functions. The exact formula of NTK with ReLU activation has been derived for feedforward neural networks (Jacot et al., 2018), convolutional neural networks (Arora et al., 2019b), and Graph Neural Networks (Du et al., 2019b).

Our theory builds upon this equivalence of network learning and kernel regression to more precisely characterize the function learned by a sufficiently-wide neural network given any specific training set. In particular, the difference between the learned function and true function over the domain of $\mathcal{X}$ determines the extrapolation error.

However, in general it is non-trivial to compute or analyze the *functional form* of what a neural network learns using Eqn. 9, because the kernel regression solution using neural tangent kernel only gives point-wise evaluation. Thus, we instead analyze the function learned by a network in the NTK's induced *feature space*, because representations in the feature space would give a functional form.

Lemma 2 makes this connection more precise: the solution to the kernel regression using neural tangent kernel, which also equals over-parameterized network learning, is equivalent to a min-norm solution among functions in the NTK's induced feature space that fits all training data. Here the min-norm refers to the RKHS norm.

**Lemma 2.** *Let $\phi(\boldsymbol{x})$ be a feature map induced by a neural tangent kernel, for any $\boldsymbol{x} \in \mathbb{R}^d$. The solution to kernel regression Eqn. 9 is equivalent to $f_{NTK}(\boldsymbol{x}) = \phi(\boldsymbol{x})^\top \beta_{NTK}$, where $\beta_{NTK}$ is*

$$\min_{\boldsymbol{\beta}} \|\boldsymbol{\beta}\|_2$$

$$s.t. \quad \phi(\boldsymbol{x}_i)^\top \boldsymbol{\beta} = y_i, \quad for \ i = 1, ..., n.$$

We prove Lemma 2 in Appendix B.6. To analyze the learned functions as the min-norm solution in feature space, we also need the explicit formula of an induced feature map of the corresponding neural tangent kernel. The following lemma gives a NTK feature space for two-layer MLPs with ReLU activation. It follows easily from the kernel formula described in Jacot et al. (2018); Arora et al. (2019b); Bietti & Mairal (2019).

**Lemma 3.** *An infinite-dimensional feature map $\phi(\boldsymbol{x})$ induced by the neural tangent kernel of a two-layer multi-layer perceptron with ReLU activation function is*

$$\phi(\boldsymbol{x}) = c\left(\boldsymbol{x} \cdot \mathbb{I}\left(\boldsymbol{w}^{(k)\top}\boldsymbol{x} \geq 0\right), \boldsymbol{w}^{(k)\top}\boldsymbol{x} \cdot \mathbb{I}\left(\boldsymbol{w}^{(k)\top}\boldsymbol{x} \geq 0\right), ...\right), \tag{10}$$

*where $\boldsymbol{w}^{(k)} \sim \mathcal{N}(\boldsymbol{0}, \boldsymbol{I})$, with $k$ going to infinity. $c$ is a constant, and $\mathbb{I}$ is the indicator function.*

We prove Lemma 3 in Appendix B.7. The feature maps for other architectures, e.g., Graph Neural Networks (GNNs) can be derived similarly. We analyze the Graph Neural Tangent Kernel (GNTK) for a simple GNN architecture in Theorem 3.

We then use Lemma 2 and 3 to characterize the properties of functions learned by an over-parameterized neural network. We precisely characterize the neural networks' learned functions in the NTK regime via solving the constrained optimization problem corresponding to the min-norm function in NTK feature space with the constraint of fitting the training data.

However, there still remains many technical challenges. For example, provable extrapolation (exact or asymptotic) is often *not* achieved with most training data distribution. Understanding the desirable condition requires significant insights into the geometry properties of training data distribution, and how they interact with the solution learned by neural networks. Our insights and refined analysis shows in $\mathbb{R}^d$ space, we need to consider the directions of training data. In graphs, we need to consider, in addition, the graph structure of training data. We refer readers to detailed proofs for the intuition of data conditions. Moreover, since NTK corresponds to infinitely wide neural networks, the feature space is of infinite dimension. The analysis of infinite dimensional spaces poses non-trivial technical challenges too.

Since different theorems have their respective challenges and insights/techniques, we refer the interested readers to the respective proofs for details. In Lemma 1 (proof in Appendix B.2), Theorem 2 (proof in Appendix B.3), and Theorem 1 (proof in Appendix B.1) we analyze over-parameterized MLPs. The proof of Corollary 1 is in Appendix B.4. In Theorem 3 we analyze Graph Neural Networks (proof in Appendix B.5).

# B PROOFS

## B.1 PROOF OF THEOREM 1

To show neural network outputs $f(\boldsymbol{x})$ converge to a linear function along all directions $\boldsymbol{v}$, we will analyze the function learned by a neural network on the training set $\{(\boldsymbol{x}_i, y_i)\}_{i=1}^n$, by studying the functional representation in the network's neural tangent kernel RKHS space.

Recall from Section A that in the NTK regime, i.e., networks are infinitely wide, randomly initialized, and trained by gradient descent with infinitesimally small learning rate, the learning dynamics of the neural network is equivalent to that of a kernel regression with respect to its neural tangent kernel.

For any $\boldsymbol{x} \in \mathbb{R}^d$, the network output is given by

$$f(\boldsymbol{x}) = \left(\langle\phi(\boldsymbol{x}), \phi(\boldsymbol{x}_1)\rangle, ..., \langle\phi(\boldsymbol{x}), \phi(\boldsymbol{x}_n)\rangle\right) \cdot \text{NTK}_{\text{train}}^{-1}\boldsymbol{Y},$$

where $\text{NTK}_{\text{train}}$ is the $n \times n$ kernel for training data, $\langle\phi(\boldsymbol{x}), \phi(\boldsymbol{x}_i)\rangle$ is the kernel value between test data $\boldsymbol{x}$ and training data $\boldsymbol{x}_i$, and $\boldsymbol{Y}$ is training labels. By Lemma 2, the kernel regression solution is also equivalent to the min-norm solution in the NTK RKHS space that fits all training data

$$f(\boldsymbol{x}) = \phi(\boldsymbol{x})^\top \boldsymbol{\beta}_{\text{NTK}}, \tag{11}$$

where the representation coefficient $\boldsymbol{\beta}_{\text{NTK}}$ is

$$\min_{\boldsymbol{\beta}} \|\boldsymbol{\beta}\|_2$$

$$\text{s.t.} \quad \phi(\boldsymbol{x}_i)^\top\boldsymbol{\beta} = y_i, \quad \text{for } i = 1, ..., n.$$

The feature map $\phi(\boldsymbol{x})$ for a two-layer MLP with ReLU activation is given by Lemma 3

$$\phi\left(\boldsymbol{x}\right) = c'\left(\boldsymbol{x}\cdot\mathbb{I}\left(\boldsymbol{w}^{(k)^\top}\boldsymbol{x}\geq 0\right), \boldsymbol{w}^{(k)^\top}\boldsymbol{x}\cdot\mathbb{I}\left(\boldsymbol{w}^{(k)^\top}\boldsymbol{x}\geq 0\right), ...\right), \tag{12}$$

where $\boldsymbol{w}^{(k)}\sim\mathcal{N}(\boldsymbol{0},\boldsymbol{I})$, with $k$ going to infinity. $c'$ is a constant, and $\mathbb{I}$ is the indicator function. Without loss of generality, we assume the bias term to be 1. For simplicity of notations, we denote each data $\boldsymbol{x}$ plus bias term by, i.e., $\hat{x} = [\boldsymbol{x}|1]$ (Bietti & Mairal, 2019), and assume constant term is 1.

Given any direction $\boldsymbol{v}$ on the unit sphere, the network outputs for out-of-distribution data $\boldsymbol{x}_0 = t\boldsymbol{v}$ and $\boldsymbol{x} = \boldsymbol{x}_0 + h\boldsymbol{v} = (1+\lambda)\boldsymbol{x}_0$, where we introduce the notation of $\boldsymbol{x}$ and $\lambda$ for convenience, are given by Eqn. 11 and Eqn. 12

$$f(\hat{\boldsymbol{x}_0}) = \boldsymbol{\beta}_{\text{NTK}}^\top\left(\hat{\boldsymbol{x}_0}\cdot\mathbb{I}\left(\boldsymbol{w}^{(k)^\top}\hat{\boldsymbol{x}_0}\geq 0\right), \boldsymbol{w}^{(k)^\top}\hat{\boldsymbol{x}_0}\cdot\mathbb{I}\left(\boldsymbol{w}^{(k)^\top}\hat{\boldsymbol{x}_0}\geq 0\right), ...\right),$$

$$f(\hat{\boldsymbol{x}}) = \boldsymbol{\beta}_{\text{NTK}}^\top\left(\hat{\boldsymbol{x}}\cdot\mathbb{I}\left(\boldsymbol{w}^{(k)^\top}\hat{\boldsymbol{x}}\geq 0\right), \boldsymbol{w}^{(k)^\top}\hat{\boldsymbol{x}}\cdot\mathbb{I}\left(\boldsymbol{w}^{(k)^\top}\hat{\boldsymbol{x}}\geq 0\right), ...\right),$$

where we have $\hat{\boldsymbol{x}_0} = [\boldsymbol{x}_0|1]$ and $\hat{\boldsymbol{x}} = [(1+\lambda)\boldsymbol{x}_0|1]$. It follows that

$$f(\hat{\boldsymbol{x}}) - f(\hat{\boldsymbol{x}_0}) = \boldsymbol{\beta}_{\text{NTK}}^\top\left(\hat{\boldsymbol{x}}\cdot\mathbb{I}\left(\boldsymbol{w}^{(k)^\top}\hat{\boldsymbol{x}}\geq 0\right) - \hat{\boldsymbol{x}_0}\cdot\mathbb{I}\left(\boldsymbol{w}^{(k)^\top}\hat{\boldsymbol{x}_0}\geq 0\right), \tag{13}$$

$$\boldsymbol{w}^{(k)^\top}\hat{\boldsymbol{x}}\cdot\mathbb{I}\left(\boldsymbol{w}^{(k)^\top}\hat{\boldsymbol{x}}\geq 0\right) - \boldsymbol{w}^{(k)^\top}\hat{\boldsymbol{x}_0}\cdot\mathbb{I}\left(\boldsymbol{w}^{(k)^\top}\hat{\boldsymbol{x}_0}\geq 0\right), ...\right) \tag{14}$$

By re-arranging the terms, we get the following equivalent form of the entries:

$$\hat{\boldsymbol{x}}\cdot\mathbb{I}\left(\boldsymbol{w}^\top\hat{\boldsymbol{x}}\geq 0\right) - \hat{\boldsymbol{x}_0}\cdot\mathbb{I}\left(\boldsymbol{w}^\top\hat{\boldsymbol{x}_0}\geq 0\right) \tag{15}$$

$$= \hat{\boldsymbol{x}}\cdot\left(\mathbb{I}\left(\boldsymbol{w}^\top\hat{\boldsymbol{x}}\geq 0\right) - \mathbb{I}\left(\boldsymbol{w}^\top\hat{\boldsymbol{x}_0}\geq 0\right) + \mathbb{I}\left(\boldsymbol{w}^\top\hat{\boldsymbol{x}_0}\geq 0\right)\right) - \hat{\boldsymbol{x}_0}\cdot\mathbb{I}\left(\boldsymbol{w}^\top\hat{\boldsymbol{x}_0}\geq 0\right) \tag{16}$$

$$= \hat{\boldsymbol{x}}\cdot\left(\mathbb{I}\left(\boldsymbol{w}^\top\hat{\boldsymbol{x}}\geq 0\right) - \mathbb{I}\left(\boldsymbol{w}^\top\hat{\boldsymbol{x}_0}\geq 0\right)\right) + (\hat{\boldsymbol{x}} - \hat{\boldsymbol{x}_0})\cdot\mathbb{I}\left(\boldsymbol{w}^\top\hat{\boldsymbol{x}_0}\geq 0\right) \tag{17}$$

$$= [\boldsymbol{x}|1]\cdot\left(\mathbb{I}\left(\boldsymbol{w}^\top\hat{\boldsymbol{x}}\geq 0\right) - \mathbb{I}\left(\boldsymbol{w}^\top\hat{\boldsymbol{x}_0}\geq 0\right)\right) + [h\boldsymbol{v}|0]\cdot\mathbb{I}\left(\boldsymbol{w}^\top\hat{\boldsymbol{x}_0}\geq 0\right) \tag{18}$$

Similarly, we have

$$\boldsymbol{w}^\top\hat{\boldsymbol{x}}\cdot\mathbb{I}\left(\boldsymbol{w}^\top\hat{\boldsymbol{x}}\geq 0\right) - \boldsymbol{w}^\top\hat{\boldsymbol{x}_0}\cdot\mathbb{I}\left(\boldsymbol{w}^\top\hat{\boldsymbol{x}_0}\geq 0\right) \tag{19}$$

$$= \boldsymbol{w}^\top\hat{\boldsymbol{x}}\cdot\left(\mathbb{I}\left(\boldsymbol{w}^\top\hat{\boldsymbol{x}}\geq 0\right) - \mathbb{I}\left(\boldsymbol{w}^\top\hat{\boldsymbol{x}_0}\geq 0\right) + \mathbb{I}\left(\boldsymbol{w}^\top\hat{\boldsymbol{x}_0}\geq 0\right)\right) - \boldsymbol{w}^\top\hat{\boldsymbol{x}_0}\cdot\mathbb{I}\left(\boldsymbol{w}^\top\hat{\boldsymbol{x}_0}\geq 0\right) \tag{20}$$

$$= \boldsymbol{w}^\top\hat{\boldsymbol{x}}\cdot\left(\mathbb{I}\left(\boldsymbol{w}^\top\hat{\boldsymbol{x}}\geq 0\right) - \mathbb{I}\left(\boldsymbol{w}^\top\hat{\boldsymbol{x}_0}\geq 0\right)\right) + \boldsymbol{w}^\top(\hat{\boldsymbol{x}} - \hat{\boldsymbol{x}_0})\cdot\mathbb{I}\left(\boldsymbol{w}^\top\hat{\boldsymbol{x}_0}\geq 0\right) \tag{21}$$

$$= \boldsymbol{w}^\top[\boldsymbol{x}|1]\cdot\left(\mathbb{I}\left(\boldsymbol{w}^\top\hat{\boldsymbol{x}}\geq 0\right) - \mathbb{I}\left(\boldsymbol{w}^\top\hat{\boldsymbol{x}_0}\geq 0\right)\right) + \boldsymbol{w}^\top[h\boldsymbol{v}|0]\cdot\mathbb{I}\left(\boldsymbol{w}^\top\hat{\boldsymbol{x}_0}\geq 0\right) \tag{22}$$

Again, let us denote the part of $\boldsymbol{\beta}_{\text{NTK}}$ corresponding to each $\boldsymbol{w}$ by $\boldsymbol{\beta}_{\boldsymbol{w}}$. Moreover, let us denote the part corresponding to Eqn. 18 by $\boldsymbol{\beta}_{\boldsymbol{w}}^1$ and the part corresponding to Eqn. 22 by $\boldsymbol{\beta}_{\boldsymbol{w}}^2$. Then we have

$$\frac{f(\hat{\boldsymbol{x}}) - f(\hat{\boldsymbol{x}_0})}{h} \tag{23}$$

$$= \int \boldsymbol{\beta}_{\boldsymbol{w}}^{1\top}[\boldsymbol{x}/h|1/h]\cdot\left(\mathbb{I}\left(\boldsymbol{w}^\top\hat{\boldsymbol{x}}\geq 0\right) - \mathbb{I}\left(\boldsymbol{w}^\top\hat{\boldsymbol{x}_0}\geq 0\right)\right)\mathrm{d}\mathbb{P}(\boldsymbol{w}) \tag{24}$$

$$+ \int \boldsymbol{\beta}_{\boldsymbol{w}}^{1\top}[\boldsymbol{v}|0]\cdot\mathbb{I}\left(\boldsymbol{w}^\top\hat{\boldsymbol{x}_0}\geq 0\right)\mathrm{d}\mathbb{P}(\boldsymbol{w}) \tag{25}$$

$$+ \int \boldsymbol{\beta}_{\boldsymbol{w}}^2\cdot\boldsymbol{w}^\top[\boldsymbol{x}/h|1/h]\cdot\left(\mathbb{I}\left(\boldsymbol{w}^\top\hat{\boldsymbol{x}}\geq 0\right) - \mathbb{I}\left(\boldsymbol{w}^\top\hat{\boldsymbol{x}_0}\geq 0\right)\right)\mathrm{d}\mathbb{P}(\boldsymbol{w}) \tag{26}$$

$$+ \int \boldsymbol{\beta}_{\boldsymbol{w}}^2\cdot\boldsymbol{w}^\top[\boldsymbol{v}|0]\cdot\mathbb{I}\left(\boldsymbol{w}^\top\hat{\boldsymbol{x}_0}\geq 0\right)\mathrm{d}\mathbb{P}(\boldsymbol{w}) \tag{27}$$

Note that all $\boldsymbol{\beta}_{\boldsymbol{w}}$ are finite constants that depend on the training data. Next, we show that as $t\to\infty$, each of the terms above converges in $O(1/\epsilon)$ to some constant coefficient $\boldsymbol{\beta}_{\boldsymbol{v}}$ that depend on the training data and the direction $\boldsymbol{v}$. Let us first consider Eqn. 25. We have

$$\int \mathbb{I}\left(\boldsymbol{w}^\top\hat{\boldsymbol{x}_0}\geq 0\right)d\mathbb{P}(\boldsymbol{w}) = \int \mathbb{I}\left(\boldsymbol{w}^\top[\boldsymbol{x}_0|1]\geq 0\right)\mathrm{d}\mathbb{P}(\boldsymbol{w}) \tag{28}$$

$$= \int \mathbb{I}\left(\boldsymbol{w}^\top[\boldsymbol{x}_0/t|1/t]\geq 0\right)\mathrm{d}\mathbb{P}(\boldsymbol{w}) \tag{29}$$

$$\to \int \mathbb{I}\left(\boldsymbol{w}^\top[\boldsymbol{v}|0]\geq 0\right)\mathrm{d}\mathbb{P}(\boldsymbol{w}) \qquad \text{as } t\to\infty \tag{30}$$

Because $\boldsymbol{\beta}_{\boldsymbol{w}}^1$ are finite constants, it follows that

$$\int {\boldsymbol{\beta}_{\boldsymbol{w}}^1}^\top [\boldsymbol{v}|0] \cdot \mathbb{I}\left(\boldsymbol{w}^\top \hat{\boldsymbol{x}}_0 \geq 0\right) \mathrm{d}\mathbb{P}(\boldsymbol{w}) \to \int {\boldsymbol{\beta}_{\boldsymbol{w}}^1}^\top [\boldsymbol{v}|0] \cdot \mathbb{I}\left(\boldsymbol{w}^\top [\boldsymbol{v}|0] \geq 0\right) \mathrm{d}\mathbb{P}(\boldsymbol{w}), \qquad (31)$$

where the right hand side is a constant that depends on training data and direction $\boldsymbol{v}$. Next, we show the convergence rate for Eqn. 31. Given error $\epsilon > 0$, because ${\boldsymbol{\beta}_{\boldsymbol{w}}^1}^\top [\boldsymbol{v}|0]$ are finite constants, we need to bound the following by $C \cdot \epsilon$ for some constant $C$,

$$|\int \mathbb{I}\left(\boldsymbol{w}^\top \hat{\boldsymbol{x}}_0 \geq 0\right) - \mathbb{I}\left(\boldsymbol{w}^\top [\boldsymbol{v}|0] \geq 0\right) \mathrm{d}\mathbb{P}(\boldsymbol{w})| \qquad (32)$$

$$= |\int \mathbb{I}\left(\boldsymbol{w}^\top [\boldsymbol{x}_0|1] \geq 0\right) - \mathbb{I}\left(\boldsymbol{w}^\top [\boldsymbol{x}_0|0] \geq 0\right) \mathrm{d}\mathbb{P}(\boldsymbol{w})| \qquad (33)$$

Observe that the two terms in Eqn. 33 represent the volume of half-(balls) that are orthogonal to vectors $[\boldsymbol{x}_0|1]$ and $[\boldsymbol{x}_0|0]$. Hence, Eqn. 33 is the volume of the non-overlapping part of the two (half)balls, which is created by rotating an angle $\theta$ along the last coordinate. By symmetry, Eqn. 33 is linear in $\theta$. Moreover, the angle $\theta = \arctan(C/t)$ for some constant $C$. Hence, it follows that

$$|\int \mathbb{I}\left(\boldsymbol{w}^\top [\boldsymbol{x}_0|1] \geq 0\right) - \mathbb{I}\left(\boldsymbol{w}^\top [\boldsymbol{x}_0|0] \geq 0\right) \mathrm{d}\mathbb{P}(\boldsymbol{w})| = C_1 \cdot \arctan(C_2/t) \qquad (34)$$

$$\leq C_1 \cdot C_2/t \qquad (35)$$

$$= O(1/t) \qquad (36)$$

In the last inequality, we used the fact that $\arctan x < x$ for $x > 0$. Hence, $O(1/t) < \epsilon$ implies $t = O(1/\epsilon)$ as desired. Next, we consider Eqn. 24.

$$\int {\boldsymbol{\beta}_{\boldsymbol{w}}^1}^\top [\boldsymbol{x}/h|1/h] \cdot \left(\mathbb{I}\left(\boldsymbol{w}^\top \hat{\boldsymbol{x}} \geq 0\right) - \mathbb{I}\left(\boldsymbol{w}^\top \hat{\boldsymbol{x}}_0 \geq 0\right)\right) \mathrm{d}\mathbb{P}(\boldsymbol{w}) \qquad (37)$$

Let us first analyze the convergence of the following:

$$|\int \mathbb{I}\left(\boldsymbol{w}^\top \hat{\boldsymbol{x}} \geq 0\right) - \mathbb{I}\left(\boldsymbol{w}^\top \hat{\boldsymbol{x}}_0 \geq 0\right) \mathrm{d}\mathbb{P}(\boldsymbol{w})| \qquad (38)$$

$$= |\int \mathbb{I}\left(\boldsymbol{w}^\top [(1+\lambda)\boldsymbol{x}_0|1] \geq 0\right) - \mathbb{I}\left(\boldsymbol{w}^\top [\boldsymbol{x}_0|1] \geq 0\right) \mathrm{d}\mathbb{P}(\boldsymbol{w})\mathrm{d}\mathbb{P}(\boldsymbol{w})| \qquad (39)$$

$$= |\int \mathbb{I}\left(\boldsymbol{w}^\top [\boldsymbol{x}_0|\frac{1}{1+\lambda}] \geq 0\right) - \mathbb{I}\left(\boldsymbol{w}^\top [\boldsymbol{x}_0|1] \geq 0\right) \mathrm{d}\mathbb{P}(\boldsymbol{w})\mathrm{d}\mathbb{P}(\boldsymbol{w})| \to 0 \qquad (40)$$

The convergence to $0$ follows from Eqn. 34. Now we consider the convergence rate. The angle $\theta$ is at most $1 - \frac{1}{1+\lambda}$ times of that in Eqn. 34. Hence, the rate is as follows

$$\left(1 - \frac{1}{1+\lambda}\right) \cdot O\left(\frac{1}{t}\right) = \frac{\lambda}{1+\lambda} \cdot O\left(\frac{1}{t}\right) = \frac{h/t}{1+h/t} \cdot O\left(\frac{1}{t}\right) = O\left(\frac{h}{(h+t)t}\right) \qquad (41)$$

Now we get back to Eqn. 24, which simplifies as the following.

$$\int {\boldsymbol{\beta}_{\boldsymbol{w}}^1}^\top \left[\boldsymbol{v} + \frac{t\boldsymbol{v}}{h}|\frac{1}{h}\right] \cdot \left(\mathbb{I}\left(\boldsymbol{w}^\top \hat{\boldsymbol{x}} \geq 0\right) - \mathbb{I}\left(\boldsymbol{w}^\top \hat{\boldsymbol{x}}_0 \geq 0\right)\right) \mathrm{d}\mathbb{P}(\boldsymbol{w}) \qquad (42)$$

We compare the rate of growth of left hand side and the rate of decrease of right hand side (indicators).

$$\frac{t}{h} \cdot \frac{h}{(h+t)t} = \frac{1}{h+t} \to 0 \quad \text{as } t \to \infty \qquad (43)$$

$$\frac{1}{h} \cdot \frac{h}{(h+t)t} = \frac{1}{(h+t)t} \to 0 \quad \text{as } t \to \infty \qquad (44)$$

Hence, the indicators decrease faster, and it follows that Eqn. 24 converges to $0$ with rate $O(\frac{1}{\epsilon})$. Moreover, we can bound $\boldsymbol{w}$ with standard concentration techniques. Then the proofs for Eqn. 26 and Eqn. 27 follow similarly. This completes the proof.

### B.2 PROOF OF LEMMA 1

**Overview of proof.** To prove exact extrapolation given the conditions on training data, we analyze the function learned by the neural network in a functional form. The network's learned function can be precisely characterized by a solution in the network's neural tangent kernel feature space which has a minimum RKHS norm among functions that can fit all training data, i.e., it corresponds to the optimum of a constrained optimization problem. We show that the global optimum of this constrained optimization problem, given the conditions on training data, is precisely the same function as the underlying true function.

**Setup and preparation.** Let $X = \{x_1, ..., x_n\}$ and $Y = \{y_1, ..., y_n\}$ denote the training set input features and their labels. Let $\beta_g \in \mathbb{R}^d$ denote the true parameters/weights for the underlying linear function $g$, i.e.,

$$g(x) = \beta_g^\top x \quad \text{for all } x \in \mathbb{R}^d$$

Recall from Section A that in the NTK regime, where networks are infinitely wide, randomly initialized, and trained by gradient descent with infinitesimally small learning rate, the learning dynamics of a neural network is equivalent to that of a kernel regression with respect to its neural tangent kernel. Moreover, Lemma 2 tells us that this kernel regression solution can be expressed in the functional form in the neural tangent kernel's feature space. That is, the function learned by the neural network (in the ntk regime) can be precisely characterized as

$$f(x) = \phi(x)^\top \beta_{\text{NTK}},$$

where the representation coefficient $\beta_{\text{NTK}}$ is

$$\min_{\beta} \|\beta\|_2 \tag{45}$$

$$\text{s.t.} \quad \phi(x_i)^\top \beta = y_i, \quad \text{for } i = 1, ..., n. \tag{46}$$

An infinite-dimensional feature map $\phi(x)$ for a two-layer ReLU network is described in Lemma 3

$$\phi(x) = c' \left( x \cdot \mathbb{I} \left( w^{(k)^\top} x \geq 0 \right), w^{(k)^\top} x \cdot \mathbb{I} \left( w^{(k)^\top} x \geq 0 \right), ... \right),$$

where $w^{(k)} \sim \mathcal{N}(0, I)$, with $k$ going to infinity. $c'$ is a constant, and $\mathbb{I}$ is the indicator function. That is, there are infinitely many directions $w$ with Gaussian density, and each direction comes with two features. Without loss of generality, we can assume the scaling constant to be $1$.

**Constrained optimization in NTK feature space.** The representation or weight of the neural network's learned function in the neural tangent kernel feature space, $\beta_{\text{NTK}}$, consists of weight vectors for each $x \cdot \mathbb{I} \left( w^{(k)^\top} x \geq 0 \right) \in \mathbb{R}^d$ and $w^{(k)^\top} x \cdot \mathbb{I} \left( w^{(k)^\top} x \geq 0 \right) \in \mathbb{R}$. For simplicity of notation, we will use $w$ to refer to a particular $w$, without considering the index $(k)$, which does not matter for our purposes. For any $w \in \mathbb{R}^d$, we denote by $\hat{\beta}_w = (\hat{\beta}_w^{(1)}, ..., \hat{\beta}_w^{(d)}) \in \mathbb{R}^d$ the weight vectors corresponding to $x \cdot \mathbb{I} \left( w^\top x \geq 0 \right)$, and denote by $\hat{\beta}_w' \in \mathbb{R}^d$ the weight for $w^\top x \cdot \mathbb{I} \left( w^\top x \geq 0 \right)$.

Observe that for any $w \sim \mathcal{N}(0, I) \in \mathbb{R}^d$, any other vectors in the same direction will activate the same set of $x_i \in \mathbb{R}^d$. That is, if $w^\top x_i \geq 0$ for any $w \in \mathbb{R}^d$, then $(k \cdot w)^\top x_i \geq 0$ for any $k > 0$. Hence, we can reload our notation to combine the effect of weights for $w$'s in the same direction. This enables simpler notations and allows us to change the distribution of $w$ in NTK features from Gaussian distribution to uniform distribution on the unit sphere.

More precisely, we reload our notation by using $\beta_w$ and $\beta_w'$ to denote the combined effect of all weights $(\hat{\beta}_{kw}^{(1)}, ..., \hat{\beta}_{kw}^{(d)}) \in \mathbb{R}^d$ and $\hat{\beta}_{kw}' \in \mathbb{R}$ for all $kw$ with $k > 0$ in the same direction of $w$. That is, for each $w \sim \text{Uni(unit sphere)} \in \mathbb{R}^d$, we define $\beta_w^{(j)}$ as the total effect of weights in the same direction

$$\beta_w^{(j)} = \int \hat{\beta}_u^{(j)} \mathbb{I} \left( \frac{w^\top u}{\|w\| \cdot \|u\|} = 1 \right) d\mathbb{P}(u), \quad \text{for } j = [d] \tag{47}$$

where $\boldsymbol{u} \sim \mathcal{N}(\boldsymbol{0}, \boldsymbol{I})$. Note that to ensure the $\boldsymbol{\beta_w}$ is a well-defined number, here we can work with the polar representation and integrate with respect to an angle. Then $\boldsymbol{\beta_w}$ is well-defined. But for simplicity of exposition, we use the plain notation of integral. Similarly, we define $\boldsymbol{\beta_w'}$ as reloading the notation of

$$\boldsymbol{\beta_w'} = \int \hat{\boldsymbol{\beta}}_{\boldsymbol{u}} \mathbb{I}\left(\frac{\boldsymbol{w}^\top \boldsymbol{u}}{\|\boldsymbol{w}\| \cdot \|\boldsymbol{u}\|} = 1\right) \cdot \frac{\|\boldsymbol{u}\|}{\|\boldsymbol{w}\|} \mathrm{d}\mathbb{P}(\boldsymbol{u}) \tag{48}$$

Here, in Eqn. 48 we have an extra term of $\frac{\|\boldsymbol{u}\|}{\|\boldsymbol{w}\|}$ compared to Eqn. 47 because the NTK features that Eqn. 48 corresponds to, $\boldsymbol{w}^\top \boldsymbol{x} \cdot \mathbb{I}\left(\boldsymbol{w}^\top \boldsymbol{x} \geq 0\right)$, has an extra $\boldsymbol{w}^\top$ term. So we need to take into account the scaling. This abstraction enables us to make claims on the high-level parameters $\boldsymbol{\beta_w}$ and $\boldsymbol{\beta_w'}$ only, which we will show to be sufficient to determine the learned function.

Then we can formulate the constrained optimization problem whose solution gives a functional form of the neural network's learned function. We rewrite the min-norm solution in Eqn. 45 as

$$\min_{\boldsymbol{\beta}} \int \left(\boldsymbol{\beta_w^{(1)}}\right)^2 + \left(\boldsymbol{\beta_w^{(2)}}\right)^2 + ... + \left(\boldsymbol{\beta_w^{(d)}}\right)^2 + (\boldsymbol{\beta_w'})^2 \, \mathrm{d}\mathbb{P}(\boldsymbol{w}) \tag{49}$$

$$\text{s.t.} \int_{\boldsymbol{w}^\top \boldsymbol{x}_i \geq 0} \boldsymbol{\beta_w^\top} \boldsymbol{x}_i + \boldsymbol{\beta_w'} \cdot \boldsymbol{w}^\top \boldsymbol{x}_i \, \mathrm{d}\mathbb{P}(\boldsymbol{w}) = \boldsymbol{\beta_g^\top} \boldsymbol{x}_i \quad \forall i \in [n], \tag{50}$$

where the density of $\boldsymbol{w}$ is now uniform on the unit sphere of $\mathbb{R}^d$. Observe that since $\boldsymbol{w}$ is from a uniform distribution, the probability density function $\mathbb{P}(\boldsymbol{w})$ is a constant. This means every $\boldsymbol{x}_i$ is activated by half of the $\boldsymbol{w}$ on the unit sphere, which implies we can now write the right hand side of Eqn. 50 in the form of left hand side, i.e., integral form. This allows us to further simplify Eqn. 50 as

$$\int_{\boldsymbol{w}^\top \boldsymbol{x}_i \geq 0} \left(\boldsymbol{\beta_w^\top} + \boldsymbol{\beta_w'} \cdot \boldsymbol{w}^\top - 2 \cdot \boldsymbol{\beta_g^\top}\right) \boldsymbol{x}_i \, \mathrm{d}\mathbb{P}(\boldsymbol{w}) = 0 \quad \forall i \in [n], \tag{51}$$

where Eqn. 51 follows from the following steps of simplification

$$\int_{\boldsymbol{w}^\top \boldsymbol{x}_i \geq 0} \boldsymbol{\beta_w^{(1)}} \boldsymbol{x}_i^{(1)} + ..\boldsymbol{\beta_w^{(d)}} \boldsymbol{x}_i^{(d)} + \boldsymbol{\beta_w'} \cdot \boldsymbol{w}^\top \boldsymbol{x}_i \mathrm{d}\mathbb{P}(\boldsymbol{w}) = \boldsymbol{\beta_g^{(1)}} \boldsymbol{x}_i^{(1)} + ...\boldsymbol{\beta_g^{(d)}} \boldsymbol{x}_i^{(d)} \quad \forall i \in [n],$$

$$\iff \int_{\boldsymbol{w}^\top \boldsymbol{x}_i \geq 0} \boldsymbol{\beta_w^{(1)}} \boldsymbol{x}_i^{(1)} + ... + \boldsymbol{\beta_w^{(d)}} \boldsymbol{x}_i^{(d)} + \boldsymbol{\beta_w'} \cdot \boldsymbol{w}^\top \boldsymbol{x}_i \, \mathrm{d}\mathbb{P}(\boldsymbol{w})$$

$$= \frac{1}{\int_{\boldsymbol{w}^\top \boldsymbol{x}_i \geq 0} \mathrm{d}\mathbb{P}(\boldsymbol{w})} \cdot \int_{\boldsymbol{w}^\top \boldsymbol{x}_i \geq 0} \mathrm{d}\mathbb{P}(\boldsymbol{w}) \cdot \left(\boldsymbol{\beta_g^{(1)}} \boldsymbol{x}_i^{(1)} + ... + \boldsymbol{\beta_g^{(d)}} \boldsymbol{x}_i^{(d)}\right) \quad \forall i \in [n],$$

$$\iff \int_{\boldsymbol{w}^\top \boldsymbol{x}_i \geq 0} \boldsymbol{\beta_w^{(1)}} \boldsymbol{x}_i^{(1)} + ... + \boldsymbol{\beta_w^{(d)}} \boldsymbol{x}_i^{(d)} + \boldsymbol{\beta_w'} \cdot \boldsymbol{w}^\top \boldsymbol{x}_i \mathrm{d}\mathbb{P}(\boldsymbol{w})$$

$$= 2 \cdot \int_{\boldsymbol{w}^\top \boldsymbol{x}_i \geq 0} \boldsymbol{\beta_g^{(1)}} \boldsymbol{x}_i^{(1)} + ... + \boldsymbol{\beta_g^{(d)}} \boldsymbol{x}_i^{(d)} \mathrm{d}\mathbb{P}(\boldsymbol{w}) \quad \forall i \in [n],$$

$$\iff \int_{\boldsymbol{w}^\top \boldsymbol{x}_i \geq 0} \left(\boldsymbol{\beta_w^\top} + \boldsymbol{\beta_w'} \cdot \boldsymbol{w}^\top - 2 \cdot \boldsymbol{\beta_g^\top}\right) \boldsymbol{x}_i \, \mathrm{d}\mathbb{P}(\boldsymbol{w}) = 0 \quad \forall i \in [n].$$

**Claim 1.** *Without loss of generality, assume the scaling factor $c$ in NTK feature map $\phi(\boldsymbol{x})$ is 1. Then the global optimum to the constraint optimization problem Eqn. 49 subject to Eqn. 51, i.e.,*

$$\min_{\boldsymbol{\beta}} \int \left(\boldsymbol{\beta_w^{(1)}}\right)^2 + \left(\boldsymbol{\beta_w^{(2)}}\right)^2 + ... + \left(\boldsymbol{\beta_w^{(d)}}\right)^2 + (\boldsymbol{\beta_w'})^2 \, \mathrm{d}\mathbb{P}(\boldsymbol{w}) \tag{52}$$

$$\text{s.t.} \int_{\boldsymbol{w}^\top \boldsymbol{x}_i \geq 0} \left(\boldsymbol{\beta_w^\top} + \boldsymbol{\beta_w'} \cdot \boldsymbol{w}^\top - 2 \cdot \boldsymbol{\beta_g^\top}\right) \boldsymbol{x}_i \, \mathrm{d}\mathbb{P}(\boldsymbol{w}) = 0 \quad \forall i \in [n]. \tag{53}$$

*satisfies $\boldsymbol{\beta_w} + \boldsymbol{\beta_w'} \cdot \boldsymbol{w} = 2\boldsymbol{\beta_g}$ for all $\boldsymbol{w}$.*

This claim implies the exact extrapolation we want to prove, i.e., $f_{\text{NTK}}(\boldsymbol{x}) = g(\boldsymbol{x})$. This is because, if our claim holds, then for any $\boldsymbol{x} \in \mathbb{R}^d$

$$
\begin{aligned}
f_{\text{NTK}}(\boldsymbol{x}) &= \int_{\boldsymbol{w}^\top \boldsymbol{x} \geq 0} \boldsymbol{\beta}_w^\top \boldsymbol{x} + \boldsymbol{\beta}_{\boldsymbol{w}}' \cdot \boldsymbol{w}^\top \boldsymbol{x} \ \mathrm{d}\mathbb{P}(\boldsymbol{w}) \\
&= \int_{\boldsymbol{w}^\top \boldsymbol{x} \geq 0} 2 \cdot \boldsymbol{\beta}_g^\top \boldsymbol{x} \ \mathrm{d}\mathbb{P}(\boldsymbol{w}) \\
&= \int_{\boldsymbol{w}^\top \boldsymbol{x} \geq 0} \mathrm{d}\mathbb{P}(\boldsymbol{w}) \cdot 2\boldsymbol{\beta}_g^\top \boldsymbol{x} \\
&= \frac{1}{2} \cdot 2\boldsymbol{\beta}_g^\top \boldsymbol{x} = g(\boldsymbol{x})
\end{aligned}
$$

Thus, it remains to prove Claim 1. To compute the optimum to the constrained optimization problem Eqn. 52, we consider the Lagrange multipliers. It is clear that the objective Eqn. 52 is convex. Moreover, the constraint Eqn. 53 is affine. Hence, by KKT, solution that satisfies the Lagrange condition will be the global optimum. We compute the Lagrange multiplier as

$$
\mathcal{L}(\boldsymbol{\beta}, \lambda) = \int \left(\boldsymbol{\beta}_{\boldsymbol{w}}^{(1)}\right)^2 + \left(\boldsymbol{\beta}_{\boldsymbol{w}}^{(2)}\right)^2 + ... + \left(\boldsymbol{\beta}_{\boldsymbol{w}}^{(d)}\right)^2 + (\boldsymbol{\beta}_{\boldsymbol{w}}')^2 \ \mathrm{d}\mathbb{P}(\boldsymbol{w}) \tag{54}
$$

$$
- \sum_{i=1}^n \lambda_i \cdot \left( \int_{\boldsymbol{w}^\top \boldsymbol{x}_i \geq 0} \left(\boldsymbol{\beta}_{\boldsymbol{w}}^\top + \boldsymbol{\beta}_{\boldsymbol{w}}' \cdot \boldsymbol{w}^\top - 2 \cdot \boldsymbol{\beta}_g^\top\right) \boldsymbol{x}_i \ \mathrm{d}\mathbb{P}(\boldsymbol{w}) \right) \tag{55}
$$

Setting the partial derivative of $\mathcal{L}(\boldsymbol{\beta}, \lambda)$ with respect to each variable to zero gives

$$
\frac{\partial \mathcal{L}}{\partial \boldsymbol{\beta}_{\boldsymbol{w}}^{(k)}} = 2\boldsymbol{\beta}_{\boldsymbol{w}}^{(k)} \mathbb{P}(\boldsymbol{w}) + \sum_{i=1}^n \lambda_i \cdot \boldsymbol{x}_i^{(k)} \cdot \mathbb{I}\left(\boldsymbol{w}^\top \boldsymbol{x}_i \geq 0\right) = 0 \tag{56}
$$

$$
\frac{\partial \mathcal{L}}{\boldsymbol{\beta}_{\boldsymbol{w}}'} = 2\boldsymbol{\beta}_{\boldsymbol{w}}' \mathbb{P}(\boldsymbol{w}) + \sum_{i=1}^n \lambda_i \cdot \boldsymbol{w}^\top \boldsymbol{x}_i \cdot \mathbb{I}\left(\boldsymbol{w}^\top \boldsymbol{x}_i \geq 0\right) = 0 \tag{57}
$$

$$
\frac{\partial \mathcal{L}}{\partial \lambda_i} = \int_{\boldsymbol{w}^\top \boldsymbol{x}_i \geq 0} \left(\boldsymbol{\beta}_{\boldsymbol{w}}^\top + \boldsymbol{\beta}_{\boldsymbol{w}}' \cdot \boldsymbol{w}^\top - 2 \cdot \boldsymbol{\beta}_g^\top\right) \boldsymbol{x}_i \ \mathrm{d}\mathbb{P}(\boldsymbol{w}) = 0 \tag{58}
$$

It is clear that the solution in Claim 1 immediately satisfies Eqn. 58. Hence, it remains to show there exist a set of $\lambda_i$ for $i \in [n]$ that satisfies Eqn. 56 and Eqn. 57. We can simplify Eqn. 56 as

$$
\boldsymbol{\beta}_{\boldsymbol{w}}^{(k)} = c \cdot \sum_{i=1}^n \lambda_i \cdot \boldsymbol{x}_i^{(k)} \cdot \mathbb{I}\left(\boldsymbol{w}^\top \boldsymbol{x}_i \geq 0\right), \tag{59}
$$

where $c$ is a constant. Similarly, we can simplify Eqn. 57 as

$$
\boldsymbol{\beta}_{\boldsymbol{w}}' = c \cdot \sum_{i=1}^n \lambda_i \cdot \boldsymbol{w}^\top \boldsymbol{x}_i \cdot \mathbb{I}\left(\boldsymbol{w}^\top \boldsymbol{x}_i \geq 0\right) \tag{60}
$$

Observe that combining Eqn. 59 and Eqn. 60 implies that the constraint Eqn. 60 can be further simplified as

$$
\boldsymbol{\beta}_{\boldsymbol{w}}' = \boldsymbol{w}^\top \boldsymbol{\beta}_{\boldsymbol{w}} \tag{61}
$$

It remains to show that given the condition on training data, there exists a set of $\lambda_i$ so that Eqn. 59 and Eqn. 61 are satisfied.

**Global optimum via the geometry of training data.** Recall that we assume our training data $\{(\boldsymbol{x}_i, y_i)\}_{i=1}^n$ satisfies for any $\boldsymbol{w} \in \mathbb{R}^d$, there exist $d$ linearly independent $\{\boldsymbol{x}_i^{\boldsymbol{w}}\}_{i=1}^d \subset \boldsymbol{X}$, where $\boldsymbol{X} = \{\boldsymbol{x}_i\}_{i=1}^n$, so that $\boldsymbol{w}^\top \boldsymbol{x}_i^{\boldsymbol{w}} \geq 0$ and $-\boldsymbol{x}_i^{\boldsymbol{w}} \in \boldsymbol{X}$ for $i = 1..d$, e.g., an orthogonal basis of $\mathbb{R}^d$ and their opposite vectors. We will show that under this data regime, we have

**(a)** for any particular $w$, there indeed exist a set of $\lambda_i$ that can satisfy the constraints Eqn. 59 and Eqn. 61 for this particular $w$.

**(b)** For any $w_1$ and $w_2$ that activate the exact same set of $\{x_i\}$, the same set of $\lambda_i$ can satisfy the constraints Eqn. 59 and Eqn. 61 of both $w_1$ and $w_2$.

**(c)** Whenever we rotate a $w_1$ to a $w_2$ so that the set of $x_i$ being activated changed, we can still find $\lambda_i$ that satisfy constraint of both $w_1$ and $w_2$.

Combining (a), (b) and (c) implies there exists a set of $\lambda$ that satisfy the constraints for all $w$. Hence, it remains to show these three claims.

We first prove Claim (a). For each $w$, we must find a set of $\lambda_i$ so that the following hold.

$$\beta_w^{(k)} = c \cdot \sum_{i=1}^n \lambda_i \cdot x_i^{(k)} \cdot \mathbb{I}\left(w^\top x_i \geq 0\right),$$

$$\beta_w' = w^\top \beta_w$$

$$\beta_w + \beta_w' \cdot w = 2\beta_g$$

Here, $\beta_g$ and $w$ are fixed, and $w$ is a vector on the unit sphere. It is easy to see that $\beta_w$ is then determined by $\beta_g$ and $w$, and there indeed exists a solution (solving a consistent linear system). Hence we are left with a linear system with $d$ linear equations

$$\beta_w^{(k)} = c \cdot \sum_{i=1}^n \lambda_i \cdot x_i^{(k)} \cdot \mathbb{I}\left(w^\top x_i \geq 0\right) \quad \forall k \in [d]$$

to solve with free variables being $\lambda_i$ so that $w$ activates $x_i$, i.e., $w^\top x_i \geq 0$. Because the training data $\{(x_i, y_i)\}_{i=1}^n$ satisfies for any $w$, there exist at least $d$ linearly independent $x_i$ that activate $w$. This guarantees for any $w$ we must have at least $d$ free variables. It follows that there must exist solutions $\lambda_i$ to the linear system. This proves Claim (a).

Next, we show that (b) for any $w_1$ and $w_2$ that activate the exact same set of $\{x_i\}$, the same set of $\lambda_i$ can satisfy the constraints Eqn. 59 and Eqn. 61 of both $w_1$ and $w_2$. Because $w_1$ and $w_2$ are activated by the same set of $x_i$, this implies

$$\beta_{w_1} = c \cdot \sum_{i=1}^n \lambda_i \cdot x_i \cdot \mathbb{I}\left(w_1^\top x_i \geq 0\right) = c \cdot \sum_{i=1}^n \lambda_i \cdot x_i \cdot \mathbb{I}\left(w_2^\top x_i \geq 0\right) = \beta_{w_2}$$

Since $\lambda_i$ already satisfy constraint Eqn. 59 for $w_1$, they also satisfy that for $w_2$. Thus, it remains to show that $\beta_{w_1} + \beta_{w_1}' \cdot w_1 = \beta_{w_2} + \beta_{w_2}' \cdot w_1$ assuming $\beta_{w_1} = \beta_{w_2}$, $\beta_{w_1}' = w_1^\top \beta_{w_1}$, and $\beta_{w_2}' = w_2^\top \beta_{w_2}$. This indeed holds because

$$\beta_{w_1} + \beta_{w_1}' \cdot w_1 = \beta_{w_2} + \beta_{w_2}' \cdot w_2$$
$$\iff \beta_{w_1}' \cdot w_1^\top = \beta_{w_2}' \cdot w_2^\top$$
$$\iff w_1^\top \beta_{w_1} w_1^\top = w_2^\top \beta_{w_2} w_2^\top$$
$$\iff w_1^\top w_1 \beta_{w_1}^\top = w_2^\top w_2 \beta_{w_2}^\top$$
$$\iff 1 \cdot \beta_{w_1}^\top = 1 \cdot \beta_{w_2}^\top$$
$$\iff \beta_{w_1} = \beta_{w_1}$$

Here, we used the fact that $w_1$ and $w_2$ are vectors on the unit sphere. This proves Claim (b).

Finally, we show (c) that Whenever we rotate a $w_1$ to a $w_2$ so that the set of $x_i$ being activated changed, we can still find $\lambda_i$ that satisfy constraint of both $w_1$ and $w_2$. Suppose we rotate $w_1$ to $w_2$ so that $w_2$ lost activation with $x_1, x_2, ..., x_p$ which in the set of linearly independent $x_i$'s being activated by $w_1$ and their opposite vectors $-x_i$ are also in the training set (without loss of generality). Then $w_2$ must now also get activated by $-x_1, -x_2, ..., -x_p$. This is because if $w_2^\top x_i < 0$, we must have $w_2^\top (-x_i) > 0$.

Recall that in the proof of Claim (a), we only needed the $\lambda_i$ from linearly independent $x_i$ that we used to solve the linear systems, and their opposite as the free variables to solve the linear system of

$d$ equations. Hence, we can set $\lambda$ to 0 for the other $\boldsymbol{x}_i$ while still satisfying the linear system. Then, suppose there exists $\lambda_i$ that satisfy

$$\boldsymbol{\beta}_{\boldsymbol{w}_1}^{(k)} = c \cdot \sum_{i=1}^{d} \lambda_i \cdot \boldsymbol{x}_i^{(k)}$$

where the $\boldsymbol{x}_i$ are the linearly independent vectors that activate $\boldsymbol{w}_1$ with opposite vectors in the training set, which we have proved in (a). Then we can satisfy the constraint for $\boldsymbol{\beta}_{\boldsymbol{w}_2}$ below

$$\boldsymbol{\beta}_{\boldsymbol{w}_2}^{(k)} = c \cdot \sum_{i=1}^{p} \hat{\lambda}_i \cdot (-\boldsymbol{x}_i)^{(k)} + \sum_{i=p+1}^{d} \lambda_i \cdot \boldsymbol{x}_i^{(k)}$$

by setting $\hat{\lambda}_i = -\lambda_i$ for $i = 1...p$. Indeed, this gives

$$\boldsymbol{\beta}_{\boldsymbol{w}_2}^{(k)} = c \cdot \sum_{i=1}^{p} (-\lambda_i) \cdot (-\boldsymbol{x}_i)^{(k)} + \sum_{i=p+1}^{d} \lambda_i \cdot \boldsymbol{x}_i^{(k)}$$

$$= c \cdot \sum_{i=1}^{d} \lambda_i \cdot \boldsymbol{x}_i^{(k)}$$

Thus, we can also find $\lambda_i$ that satisfy the constraint for $\boldsymbol{\beta}_{\boldsymbol{w}_2}$. Here, we do not consider the case where $\boldsymbol{w}_2$ is parallel with an $\boldsymbol{x}_i$ because such $\boldsymbol{w}_2$ has measure zero. Note that we can apply this argument iteratively because the flipping the sign always works and will not create any inconsistency.

Moreover, we can show that the constraint for $\boldsymbol{\beta}'_{\boldsymbol{w}2}$ is satisfied by a similar argument as in proof of Claim (b). This follows from the fact that our construction makes $\boldsymbol{\beta}_{\boldsymbol{w}_1} = \boldsymbol{\beta}_{\boldsymbol{w}_2}$. Then we can follow the same argument as in (b) to show that $\boldsymbol{\beta}_{\boldsymbol{w}_1} + \boldsymbol{\beta}'_{\boldsymbol{w}_1} \cdot \boldsymbol{w}_1 = \boldsymbol{\beta}_{\boldsymbol{w}_2} + \boldsymbol{\beta}'_{\boldsymbol{w}_2} \cdot \boldsymbol{w}_1$. This completes the proof of Claim (c).

In summary, combining Claim (a), (b) and (c) gives that Claim 1 holds. That is, given our training data, the global optimum to the constrained optimization problem of finding the min-norm solution among functions that fit the training data satisfies $\boldsymbol{\beta}_{\boldsymbol{w}} + \boldsymbol{\beta}'_{\boldsymbol{w}} \cdot \boldsymbol{w} = 2\boldsymbol{\beta}_g$. We also showed that this claim implies exact extrapolation, i.e., the network's learned function $f(\boldsymbol{x})$ is equal to the true underlying function $g(\boldsymbol{x})$ for all $\boldsymbol{x} \in \mathbb{R}^d$. This completes the proof.

### B.3 PROOF OF THEOREM 2

Proof of the asymptotic convergence to extrapolation builds upon our proof of exact extrapolation, i.e., Lemma 1. The proof idea is that if the training data distribution has support at all directions, when the number of samples $n \to \infty$, asymptotically the training set will converge to some imaginary training set that satisfies the condition for exact extrapolation. Since if training data are close the neural tangent kernels are also close, the predictions or learned function will converge to a function that achieves perfect extrapolation, that is, the true underlying function.

**Asymptotic convergence of data sets.** We first show the training data converge to a data set that satisfies the exact extrapolation condition in Lemma 1. Suppose training data $\{\boldsymbol{x}_i\}_{i=1}^{n}$ are sampled from a distribution whose support contains a connected set $\mathcal{S}$ that intersects all directions, i.e., for any non-zero $\boldsymbol{w} \in \mathbb{R}^d$, there exists $k > 0$ so that $k\boldsymbol{w} \in \mathcal{S}$.

Let us denote by $\mathcal{S}$ the set of datasets that satisfy the condition in Lemma 1. In fact, we will use a relaxed condition in the proof of Lemma 1 (Lemma 1 in the main text uses a stricter condition for simplicity of exposition). Given a general dataset $\boldsymbol{X}$ and a dataset $\boldsymbol{S} \in \mathcal{S}$ of the same size $n$, let $\sigma(\boldsymbol{X}, \boldsymbol{S})$ denote a matching of their data points, i.e., $\sigma$ outputs a sequence of pairs

$$\sigma(\boldsymbol{X}, \boldsymbol{S})_i = (\boldsymbol{x}_i, \boldsymbol{s}_i) \quad \text{for } i \in [n]$$
$$s.t. \quad \boldsymbol{X} = \{\boldsymbol{x}_i\}_{i=1}^{n}$$
$$\boldsymbol{S} = \{\boldsymbol{s}_i\}_{i=1}^{n}$$

Let $\ell : \mathbb{R}^d \times \mathbb{R}^d \to \mathbb{R}$ be the $l2$ distance that takes in a pair of points. We then define the distance between the datasets $d(\boldsymbol{X}, \boldsymbol{S})$ as the minimum sum of $l2$ distances of their data points over all

possible matching.

$$d(\boldsymbol{X}, \boldsymbol{S}) = \begin{cases} \min_{\sigma} \sum_{i=1}^{n} \ell\left(\sigma\left(\boldsymbol{X}, \boldsymbol{S}\right)_i\right) & |\boldsymbol{X}| = |\boldsymbol{S}| = n \\ \infty & |\boldsymbol{X}| \neq |\boldsymbol{S}| \end{cases}$$

We can then define a "closest distance to perfect dataset" function $\mathcal{D}^* : \mathcal{X} \to \mathbb{R}$ which maps a dataset $\boldsymbol{X}$ to the minimum distance of $\boldsymbol{X}$ to any dataset in $\mathcal{S}$

$$\mathcal{D}^*\left(\boldsymbol{X}\right) = \min_{\boldsymbol{S} \in \mathcal{S}} d\left(\boldsymbol{X}, \boldsymbol{S}\right)$$

It is easy to see that for any dataset $\boldsymbol{X} = \{\boldsymbol{x}_i\}_{i=1}^n$, $\mathcal{D}^*\left(\boldsymbol{X}\right)$ can be bounded by the minimum of the closest distance to perfect dataset $\mathcal{D}^*$ of sub-datasets of $\boldsymbol{X}$ of size $2d$.

$$\mathcal{D}^*\left(\{\boldsymbol{x}_i\}_{i=1}^n\right) \leq \min_{k=1}^{\lfloor n/2d \rfloor} \mathcal{D}^*\left(\{\boldsymbol{x}_j\}_{j=(k-1)*2d+1}^{k*2d}\right) \tag{62}$$

This is because for any $\boldsymbol{S} \in \mathcal{S}$, and any $\boldsymbol{S} \subseteq \boldsymbol{S}'$, we must have $\boldsymbol{S}' \in \mathcal{S}$ because a dataset satisfies exact extrapolation condition as long as it contains some key points. Thus, adding more data will not hurt, i.e., for any $\boldsymbol{X}_1 \subseteq \boldsymbol{X}_2$, we always have

$$\mathcal{D}^*\left(\boldsymbol{X_1}\right) \leq \mathcal{D}^*\left(\boldsymbol{X_2}\right)$$

Now let us denote by $\boldsymbol{X}_n$ a random dataset of size $n$ where each $\boldsymbol{x}_i \in \boldsymbol{X}_n$ is sampled from the training distribution. Recall that our training data $\{\boldsymbol{x}_i\}_{i=1}^n$ are sampled from a distribution whose support contains a connected set $\mathcal{S}^*$ that intersects all directions, i.e., for any non-zero $\boldsymbol{w} \in \mathbb{R}^d$, there exists $k > 0$ so that $k\boldsymbol{w} \in \mathcal{S}^*$. It follows that for a random dataset $\boldsymbol{X}_{2d}$ of size $2d$, the probability that $\mathcal{D}^*(\boldsymbol{X}_{2d}) > \epsilon$ happens is less than 1 for any $\epsilon > 0$.

First there must exist $\boldsymbol{S}_0 = \{\boldsymbol{s}_i\}_{i=1}^{2d} \in \mathcal{S}$ of size $2d$, e.g., orthogonal basis and their opposite vectors. Observe that if we scale any $\boldsymbol{s}_i$ by $k > 0$, the resulting dataset is still in $\mathcal{S}$ by the definition of $\mathcal{S}$. We denote the set of datasets where we are allowed to scale elements of $\boldsymbol{S}_0$ by $\mathcal{S}_0$. It follows that

$$\begin{aligned}
\mathbb{P}\left(\mathcal{D}^*(\boldsymbol{X}_{2d}) > \epsilon\right) &= \mathbb{P}\left(\min_{\boldsymbol{S} \in \mathcal{S}} d\left(\boldsymbol{X}_{2d}, \boldsymbol{S}\right) > \epsilon\right) \\
&\leq \mathbb{P}\left(\min_{\boldsymbol{S} \in \mathcal{S}_0} d\left(\boldsymbol{X}_{2d}, \boldsymbol{S}\right) > \epsilon\right) \\
&= \mathbb{P}\left(\min_{\boldsymbol{S} \in \mathcal{S}_0} \min_{\sigma} \sum_{i=1}^{n} \ell\left(\sigma\left(\boldsymbol{X}_{2d}, \boldsymbol{S}\right)_i\right) > \epsilon\right) \\
&= 1 - \mathbb{P}\left(\min_{\boldsymbol{S} \in \mathcal{S}_0} \min_{\sigma} \sum_{i=1}^{n} \ell\left(\sigma\left(\boldsymbol{X}_{2d}, \boldsymbol{S}\right)_i\right) \leq \epsilon\right) \\
&\leq 1 - \mathbb{P}\left(\min_{\boldsymbol{S} \in \mathcal{S}_0} \min_{\sigma} \max_{i=1}^{n} \ell\left(\sigma\left(\boldsymbol{X}_{2d}, \boldsymbol{S}\right)_i\right) \leq \epsilon\right) \\
&\leq \delta < 1
\end{aligned}$$

where we denote the bound of $\mathbb{P}\left(\mathcal{D}^*(\boldsymbol{X}_{2d}) > \epsilon\right)$ by $\delta < 1$, and the last step follows from

$$\mathbb{P}\left(\min_{\boldsymbol{S} \in \mathcal{S}_0} \min_{\sigma} \max_{i=1}^{n} \ell\left(\sigma\left(\boldsymbol{X}_{2d}, \boldsymbol{S}\right)_i\right) \leq \epsilon\right) > 0$$

which further follows from the fact that for any $\boldsymbol{s}_i \in \mathcal{S}_0$, by the assumption on training distribution, we can always find $k > 0$ so that $k\boldsymbol{s}_i \in \mathcal{S}^*$, a connected set in the support of training distribution. By the connectivity of support $\mathcal{S}^*$, $k\boldsymbol{s}_i$ cannot be an isolated point in $\mathcal{S}^*$, so for any $\epsilon > 0$, we must have

$$\int_{\|\boldsymbol{x} - k\boldsymbol{s}_i\| \leq \epsilon, \boldsymbol{x} \in \mathcal{S}^*} f_{\boldsymbol{X}}(\boldsymbol{x}) \mathrm{d}\boldsymbol{x} > 0$$

Hence, we can now apply Eqn. 62 to bound $\mathcal{D}^*(\boldsymbol{X}_n)$. Given any $\epsilon > 0$, we have

$$
\begin{aligned}
\mathbb{P}\left(\mathcal{D}^*(\boldsymbol{X}_n) > \epsilon\right) &= 1 - \mathbb{P}\left(\mathcal{D}^*(\boldsymbol{X}_n) \leq \epsilon\right) \\
&\leq 1 - \mathbb{P}\left(\min_{k=1}^{\lfloor n/2d \rfloor} \mathcal{D}^*\left(\{\boldsymbol{x}_j\}_{j=(k-1)*2d+1}^{k*2d}\right) \leq \epsilon\right) \\
&\leq 1 - \left(1 - \prod_{k=1}^{\lfloor n/2d \rfloor} \mathbb{P}\left(\mathcal{D}^*\left(\{\boldsymbol{x}_j\}_{j=(k-1)*2d+1}^{k*2d}\right) > \epsilon\right)\right) \\
&= \prod_{k=1}^{\lfloor n/2d \rfloor} \mathbb{P}\left(\mathcal{D}^*\left(\{\boldsymbol{x}_j\}_{j=(k-1)*2d+1}^{k*2d}\right) > \epsilon\right) \\
&\leq \delta^{\lfloor n/2d \rfloor}
\end{aligned}
$$

Here $\delta < 1$. This implies $\mathcal{D}^*(\boldsymbol{X}_n) \overset{p}{\longrightarrow} 0$, i.e.,

$$
\lim_{n\to\infty} \mathbb{P}\left(\mathcal{D}^*(\boldsymbol{X}_n) > \epsilon\right) = 0 \quad \forall \epsilon > 0 \tag{63}
$$

Eqn. 63 says as the number of training samples $n \to \infty$, our training set will converge in probability to a dataset that satisfies the requirement for exact extrapolation.

**Asymptotic convergence of predictions.** Let $\text{NTK}(\boldsymbol{x}, \boldsymbol{x}') : \mathbb{R}^d \times \mathbb{R}^d \to \mathbb{R}$ denote the neural tangent kernel for a two-layer ReLU MLP. It is easy to see that if $\boldsymbol{x} \to \boldsymbol{x}^*$, then $\text{NTK}(\boldsymbol{x}, \cdot) \to \text{NTK}(\boldsymbol{x}^*, \cdot)$ (Arora et al. (2019b)). Let $\text{NTK}_{\text{train}}$ denote the $n \times n$ kernel matrix for training data.

We have shown that our training set converges to a perfect data set that satisfies conditions of exact extrapolation. Moreover, note that our training set will only have a finite number of (not increase with $n$) $\boldsymbol{x}_i$ that are not precisely the same as those in a perfect dataset. This is because a perfect data only contains a finite number of key points and the other points can be replaced by any other points while still being a perfect data set. Thus, we have $\text{NTK}_{\text{train}} \to N^*$, where $N^*$ is the $n \times n$ NTK matrix for some perfect data set.

Because neural tangent kernel is positive definite, we have $\text{NTK}_{\text{train}}^{-1} \to {N^*}^{-1}$. Recall that for any $\boldsymbol{x} \in \mathbb{R}^d$, the prediction of NTK is

$$
f_{\text{NTK}}(\boldsymbol{x}) = (\text{NTK}(\boldsymbol{x}, \boldsymbol{x}_1), ..., \text{NTK}(\boldsymbol{x}, \boldsymbol{x}_n)) \cdot \text{NTK}_{\text{train}}^{-1} \boldsymbol{Y},
$$

where $\text{NTK}_{\text{train}}$ is the $n \times n$ kernel for training data, $\text{NTK}(\boldsymbol{x}, \boldsymbol{x}_i)$ is the kernel value between test data $\boldsymbol{x}$ and training data $\boldsymbol{x}_i$, and $\boldsymbol{Y}$ is training labels.

Similarly, we have $(\text{NTK}(\boldsymbol{x}, \boldsymbol{x}_1), ..., \text{NTK}(\boldsymbol{x}, \boldsymbol{x}_n)) \to (\text{NTK}(\boldsymbol{x}, \boldsymbol{x}_1^*), ..., \text{NTK}(\boldsymbol{x}, \boldsymbol{x}_n^*))$, where $x_i^*$ is a perfect data set that our training set converges to. Combining this with $\text{NTK}_{\text{train}}^{-1} \to {N^*}^{-1}$ gives

$$
f_{\text{NTK}} \overset{p}{\longrightarrow} f_{\text{NTK}}^* = g,
$$

where $f_{\text{NTK}}$ is the function learned using our training set, and $f_{\text{NTK}}^*$ is that learned using a perfect data set, which is equal to the true underlying function $g$. This completes the proof.

## B.4 Proof of Corollary 1

In order for GNN with linear aggregations

$$
h_u^{(k)} = \sum_{v \in \mathcal{N}(u)} \text{MLP}^{(k)}\left(h_u^{(k)}, h_v^{(k)}, \boldsymbol{x}_{(u,v)}\right),
$$

$$
h_G = \text{MLP}^{(K+1)}\left(\sum_{u \in G} h_u^{(K)}\right),
$$

to extrapolate in the maximum degree task, it must be able to simulate the underlying function

$$
h_G = \max_{u \in G} \sum_{v \in \mathcal{N}(u)} 1
$$

Because the max function cannot be decomposed as the composition of piece-wise linear functions, the MLP$^{(K+1)}$ module in GNN must learn a function that is not piece-wise linear over domains outside the training data range. Since Theorem 1 proves for two-layer overparameterized MLPs, here we also assume MLP$^{(K+1)}$ is a two-layer overparameterized MLP, although the result can be extended to more layers. It then follows from Theorem 1 that for any input and label (and thus gradient), MLP$^{(K+1)}$ will converge to linear functions along directions from the origin. Hence, there are always domains where the GNN cannot learn a correct target function.

## B.5 PROOF OF THEOREM 3

Our proof applies the similar proof techniques for Lemma 1 and 2 to Graph Neural Networks (GNNs). This is essentially an analysis of Graph Neural Tangent Kernel (GNTK), i.e., neural tangent kernel of GNNs.

We first define the simple GNN architecture we will be analyzing, and then present the GNTK for this architecture. Suppose $G = (V, E)$ is an input graph without edge feature, and $\boldsymbol{x}_u \in \mathbb{R}^d$ is the node feature of any node $u \in V$. Let us consider the simple one-layer GNN whose input is $G$ and output is $h_G$

$$h_G = W^{(2)} \max_{u \in G} \sum_{v \in \mathcal{N}(u)} W^{(1)} \boldsymbol{x}_v \tag{64}$$

Note that our analysis can be extended to other variants of GNNs, e.g., with non-empty edge features, ReLU activation, different neighbor aggregation and graph-level pooling architectures. We analyze this GNN for simplicity of exposition.

Next, let us calculate the feature map of the neural tangent kernel for this GNN. Recall from Section A that consider a graph neural network $f(\boldsymbol{\theta}, G) : \mathcal{G} \rightarrow \mathbb{R}$ where $\boldsymbol{\theta} \in \mathbb{R}^m$ is the parameters in the network and $G \in \mathcal{G}$ is the input graph. Then the neural tangent kernel is

$$\boldsymbol{H}_{ij} = \left\langle \frac{\partial f(\boldsymbol{\theta}, G_i)}{\partial \boldsymbol{\theta}}, \frac{\partial f(\boldsymbol{\theta}, G_j)}{\partial \boldsymbol{\theta}} \right\rangle,$$

where $\boldsymbol{\theta}$ are the infinite-dimensional parameters. Hence, the gradients with respect to all parameters give a natural feature map. Let us denote, for any node $u$, the degree of $u$ by

$$\boldsymbol{h}_u = \sum_{v \in \mathcal{N}(u)} \boldsymbol{x}_v \tag{65}$$

It then follows from simple computation of derivative that the following is a feature map of the GNTK for Eqn. 64

$$\phi(G) = c \cdot \left( \max_{u \in G} \left( \boldsymbol{w}^{(k)\top} \boldsymbol{h}_u \right), \sum_{u \in G} \mathbb{I} \left( u = \arg\max_{v \in G} \boldsymbol{w}^{(k)\top} \boldsymbol{h}_v \right) \cdot \boldsymbol{h}_u, ... \right), \tag{66}$$

where $\boldsymbol{w}^{(k)} \sim \mathcal{N}(\boldsymbol{0}, \boldsymbol{I})$, with $k$ going to infinity. $c$ is a constant, and $\mathbb{I}$ is the indicator function.

Next, given training data $\{(G_i, y_i)\}_{i=1}^n$, let us analyze the function learned by GNN through the min-norm solution in the GNTK feature space. The same proof technique is also used in Lemma 1 and 2.

Recall the assumption that all graphs have uniform node feature, i.e., the learning task only considers graph structure, but not node feature. We assume $\boldsymbol{x}_v = 1$ without loss of generality. Observe that in this case, there are two directions, positive or negative, for one-dimensional Gaussian distribution. Hence, we can simplify our analysis by combining the effect of linear coefficients for $\boldsymbol{w}$ in the same direction as in Lemma 1 and 2.

Similarly, for any $\boldsymbol{w}$, let us define $\hat{\boldsymbol{\beta}}_{\boldsymbol{w}} \in \mathbb{R}$ as the linear coefficient corresponding to $\sum_{u \in G} \mathbb{I} \left( u = \arg\max_{v \in G} \boldsymbol{w}^\top \boldsymbol{h}_v \right) \cdot \boldsymbol{h}_u$ in RKHS space, and denote by $\hat{\boldsymbol{\beta}}'_{\boldsymbol{w}} \in \mathbb{R}$ the weight for $\max_{u \in G} \left( \boldsymbol{w}^\top \boldsymbol{h}_u \right)$. Similarly, we can combine the effect of all $\hat{\boldsymbol{\beta}}$ in the same direction as in Lemma 1

and 2. We define the combined effect with $\boldsymbol{\beta_w}$ and $\boldsymbol{\beta'_w}$. This allows us to reason about $\boldsymbol{w}$ with two directions, $+$ and $-$.

Recall that the underlying reasoning function, maximum degree, is

$$g(G) = \max_{u \in G} \boldsymbol{h}_u.$$

We formulate the constrained optimization problem, i.e., min-norm solution in GNTK feature space that fits all training data, as

$$\min_{\hat{\beta}, \hat{\beta}'} \int \hat{\boldsymbol{\beta}}_{\boldsymbol{w}}^2 + \hat{\boldsymbol{\beta}}_{\boldsymbol{w}}'^2 \mathrm{d}\mathbb{P}(\boldsymbol{w})$$

$$s.t. \int \sum_{u \in G_i} \mathbb{I}\left( u = \arg\max_{v \in G} \boldsymbol{w} \cdot \boldsymbol{h}_v \right) \cdot \hat{\boldsymbol{\beta}}_{\boldsymbol{w}} \cdot \boldsymbol{h}_u + \max_{u \in G_i}(\boldsymbol{w} \cdot \boldsymbol{h}_u) \cdot \hat{\boldsymbol{\beta}}'_{\boldsymbol{w}} \mathrm{d}\mathbb{P}(\boldsymbol{w}) = \max_{u \in G_i} \boldsymbol{h}_u \quad \forall i \in [n],$$

where $G_i$ is the i-th training graph and $\boldsymbol{w} \sim \mathcal{N}(0,1)$. By combining the effect of $\hat{\boldsymbol{\beta}}$, and taking the derivative of the Lagrange for the constrained optimization problem and setting to zero, we get the global optimum solution satisfy the following constraints.

$$\boldsymbol{\beta}_+ = c \cdot \sum_{i=1}^n \lambda_i \cdot \sum_{u \in G_i} \boldsymbol{h}_u \cdot \mathbb{I}\left( u = \arg\max_{v \in G_i} \boldsymbol{h}_v \right) \tag{67}$$

$$\boldsymbol{\beta}_- = c \cdot \sum_{i=1}^n \lambda_i \cdot \sum_{u \in G_i} \boldsymbol{h}_u \cdot \mathbb{I}\left( u = \arg\min_{v \in G_i} \boldsymbol{h}_v \right) \tag{68}$$

$$\boldsymbol{\beta}'_+ = c \cdot \sum_{i=1}^n \lambda_i \cdot \max_{u \in G_i} \boldsymbol{h}_u \tag{69}$$

$$\boldsymbol{\beta}'_- = c \cdot \sum_{i=1}^n \lambda_i \cdot \min_{u \in G_i} \boldsymbol{h}_u \tag{70}$$

$$\max_{u \in G_i} \boldsymbol{h}_u = \boldsymbol{\beta}_+ \cdot \sum_{u \in G_i} \mathbb{I}\left( u = \arg\max_{v \in G_i} \boldsymbol{h}_v \right) \cdot \boldsymbol{h}_u + \boldsymbol{\beta}'_+ \cdot \max_{u \in G_i} \boldsymbol{h}_u \tag{71}$$

$$+ \boldsymbol{\beta}_- \cdot \sum_{u \in G_i} \mathbb{I}\left( u = \arg\min_{v \in G_i} \boldsymbol{h}_v \right) \cdot \boldsymbol{h}_u + \boldsymbol{\beta}'_- \cdot \min_{u \in G_i} \boldsymbol{h}_u \quad \forall i \in [n] \tag{72}$$

where $c$ is some constant, $\lambda_i$ are the Lagrange parameters. Note that here we used the fact that there are two directions $+1$ and $-1$. This enables the simplification of Lagrange derivative. For a similar step-by-step derivation of Lagrange, refer to the proof of Lemma 1.

Let us consider the solution $\boldsymbol{\beta}'_+ = 1$ and $\boldsymbol{\beta}_+ = \boldsymbol{\beta}_- = \boldsymbol{\beta}'_- = 0$. It is clear that this solution can fit the training data, and thus satisfies Eqn. 71. Moreover, this solution is equivalent to the underlying reasoning function, maximum degree, $g(G) = \max_{u \in G} \boldsymbol{h}_u$.

Hence, it remains to show that, given our training data, there exist $\lambda_i$ so that the remaining four constraints are satisfies for this solution. Let us rewrite these constraints as a linear systems where the variables are $\lambda_i$

$$\begin{pmatrix} \boldsymbol{\beta}_+ \\ \boldsymbol{\beta}_- \\ \boldsymbol{\beta}'_+ \\ \boldsymbol{\beta}'_- \end{pmatrix} = c \cdot \sum_{i=1}^n \lambda_i \cdot \begin{pmatrix} \sum_{u \in G_i} \boldsymbol{h}_u \cdot \mathbb{I}\left( u = \arg\max_{v \in G_i} \boldsymbol{h}_v \right) \\ \sum_{u \in G_i} \boldsymbol{h}_u \cdot \mathbb{I}\left( u = \arg\min_{v \in G_i} \boldsymbol{h}_v \right) \\ \max_{u \in G_i} \boldsymbol{h}_u \\ \min_{u \in G_i} \boldsymbol{h}_u \end{pmatrix} \tag{73}$$

By standard theory of linear systems, there exist $\lambda_i$ to solve Eqn. 73 if there are at least four training data $G_i$ whose following vectors linear independent

$$\begin{pmatrix} \sum_{u \in G_i} \boldsymbol{h}_u \cdot \mathbb{I}\left(u = \arg\max_{v \in G_i} \boldsymbol{h}_v\right) \\ \sum_{u \in G_i} \boldsymbol{h}_u \cdot \mathbb{I}\left(u = \arg\min_{v \in G_i} \boldsymbol{h}_v\right) \\ \max_{u \in G_i} \boldsymbol{h}_u \\ \min_{u \in G_i} \boldsymbol{h}_u \end{pmatrix} = \begin{pmatrix} \max_{u \in G_i} \boldsymbol{h}_u \cdot N_i^{\max} \\ \min_{u \in G_i} \boldsymbol{h}_u \cdot N_i^{\min} \\ \max_{u \in G_i} \boldsymbol{h}_u \\ \min_{u \in G_i} \boldsymbol{h}_u \end{pmatrix} \quad (74)$$

Here, $N_i^{\max}$ denotes the number of nodes that achieve the maximum degree in the graph $G_i$, and $N_i^{\min}$ denotes the number of nodes that achieve the min degree in the graph $G_i$. By the assumption of our training data that there are at least four $G_i \sim \mathcal{G}$ with linearly independent Eqn. 74. Hence, our simple GNN learns the underlying function as desired.

This completes the proof.

## B.6 PROOF OF LEMMA 2

Let $W$ denote the span of the feature maps of training data $\boldsymbol{x}_i$, i.e.

$$W = \text{span}\left(\phi\left(\boldsymbol{x}_1\right), \phi\left(\boldsymbol{x}_2\right), ..., \phi\left(\boldsymbol{x}_n\right)\right).$$

Then we can decompose the coordinates of $f_{\text{NTK}}$ in the RKHS space, $\boldsymbol{\beta}_{\text{NTK}}$, into a vector $\boldsymbol{\beta}_0$ for the component of $f_{\text{NTK}}$ in the span of training data features $W$, and a vector $\boldsymbol{\beta}_1$ for the component in the orthogonal complement $W^\top$, i.e.,

$$\boldsymbol{\beta}_{\text{NTK}} = \boldsymbol{\beta}_0 + \boldsymbol{\beta}_1.$$

First, note that since $f_{\text{NTK}}$ must be able to fit the training data (NTK is a universal kernel as we will discuss next), i.e.,

$$\phi(\boldsymbol{x}_i)^\top \boldsymbol{\beta}_{\text{NTK}} = y_i.$$

Thus, we have $\phi(\boldsymbol{x}_i)^\top \boldsymbol{\beta}_0 = y_i$. Then, $\boldsymbol{\beta}_0$ is uniquely determined by the kernel regression solution with respect to the neural tangent kernel

$$f_{\text{NTK}}(\boldsymbol{x}) = \left(\langle \phi(\boldsymbol{x}), \phi(\boldsymbol{x}_1)\rangle, ..., \langle \phi(\boldsymbol{x}), \phi(\boldsymbol{x}_n)\rangle\right) \cdot \text{NTK}_{\text{train}}^{-1} \boldsymbol{Y},$$

where $\text{NTK}_{\text{train}}$ is the $n \times n$ kernel for training data, $\langle \phi(\boldsymbol{x}), \phi(\boldsymbol{x}_i)\rangle$ is the kernel between test data $\boldsymbol{x}$ and training data $\boldsymbol{x}_i$, and $\boldsymbol{Y}$ is training labels.

The kernel regression solution $f_{\text{NTK}}$ is uniquely determined because the neural tangent kernel $\text{NTK}_{\text{train}}$ is positive definite assuming no two training data are parallel, which can be enforced with a bias term (Du et al., 2019c). In any case, the solution is a min-norm by pseudo-inverse.

Moreover, a unique kernel regression solution $f_{\text{NTK}}$ that spans the training data features corresponds to a unique representation in the RKHS space $\boldsymbol{\beta}_0$.

Since $\boldsymbol{\beta}_0$ and $\boldsymbol{\beta}_1$ are orthogonal, we also have the following

$$\|\boldsymbol{\beta}_{\text{NTK}}\|_2^2 = \|\boldsymbol{\beta}_0 + \boldsymbol{\beta}_1\|_2^2 = \|\boldsymbol{\beta}_0\|_2^2 + \|\boldsymbol{\beta}_1\|_2^2.$$

This implies the norm of $\boldsymbol{\beta}_{\text{NTK}}$ is at least as large as the norm of any $\boldsymbol{\beta}$ such that $\phi(\boldsymbol{x}_i)^\top \boldsymbol{\beta}_{\text{NTK}} = y_i$. Moreover, observe that the solution to kernel regression Eqn. 9 is in the feature span of training data, given the kernel matrix for training data is full rank.

$$f_{\text{NTK}}(\boldsymbol{x}) = \left(\langle \phi(\boldsymbol{x}), \phi(\boldsymbol{x}_1)\rangle, ..., \langle \phi(\boldsymbol{x}), \phi(\boldsymbol{x}_n)\rangle\right) \cdot \text{NTK}_{\text{train}}^{-1} \boldsymbol{Y}.$$

Since $\boldsymbol{\beta}_1$ is for the component of $f_{\text{NTK}}$ in the orthogonal complement of training data feature span, we must have $\boldsymbol{\beta}_1 = \boldsymbol{0}$. It follows that $\boldsymbol{\beta}_{\text{NTK}}$ is equivalent to

$$\min_{\boldsymbol{\beta}} \|\boldsymbol{\beta}\|_2$$

$$\text{s.t.} \quad \phi(\boldsymbol{x}_i)^\top \boldsymbol{\beta} = y_i, \quad \text{for } i = 1, ..., n.$$

as desired.

### B.7 PROOF OF LEMMA 3

We first compute the neural tangent kernel $\text{NTK}(\boldsymbol{x}, \boldsymbol{x}')$ for a two-layer multi-layer perceptron (MLP) with ReLU activation function, and then show that it can be induced by the feature space $\phi(\boldsymbol{x})$ specified in the lemma so that $\text{NTK}(\boldsymbol{x}, \boldsymbol{x}') = \langle \phi(\boldsymbol{x}), \phi(\boldsymbol{x}') \rangle$.

Recall that Jacot et al. (2018) have derived the general framework for computing the neural tangent kernel of a neural network with general architecture and activation function. This framework is also described in Arora et al. (2019b); Du et al. (2019b), which, in addition, compute the exact kernel formula for convolutional networks and Graph Neural Networks, respectively. Following the framework in Jacot et al. (2018) and substituting the general activation function $\sigma$ with ReLU gives the kernel formula for a two-layer MLP with ReLU activation. This has also been described in several previous works (Du et al., 2019c; Chizat et al., 2019; Bietti & Mairal, 2019).

Below we describe the general framework in Jacot et al. (2018) and Arora et al. (2019b). Let $\sigma$ denote the activation function. The neural tangent kernel for an $h$-layer multi-layer perceptron can be recursively defined via a dynamic programming process. Here, $\Sigma^{(i)} : \mathbb{R}^d \times \mathbb{R}^d \to \mathbb{R}$ for $i = 0...h$ is the covariance for the $i$-th layer.

$$\Sigma^{(0)}(\boldsymbol{x}, \boldsymbol{x}') = \boldsymbol{x}^\top \boldsymbol{x}',$$

$$\wedge^{(i)}(\boldsymbol{x}, \boldsymbol{x}') = \left( \begin{array}{cc} \Sigma^{(i-1)}(\boldsymbol{x}, \boldsymbol{x}) & \Sigma^{(i-1)}(\boldsymbol{x}, \boldsymbol{x}') \\ \Sigma^{(i-1)}(\boldsymbol{x}', \boldsymbol{x}) & \Sigma^{(i-1)}(\boldsymbol{x}', \boldsymbol{x}') \end{array} \right),$$

$$\Sigma^{(i)}(\boldsymbol{x}, \boldsymbol{x}') = c \cdot \mathop{\mathbb{E}}_{u,v \sim \mathcal{N}(\boldsymbol{0}, \wedge^{(i)})} [\sigma(u)\sigma(v)].$$

The derivative covariance is defined similarly:

$$\dot{\Sigma}^{(i)}(\boldsymbol{x}, \boldsymbol{x}') = c \cdot \mathop{\mathbb{E}}_{u,v \sim \mathcal{N}(\boldsymbol{0}, \wedge^{(i)})} [\dot{\sigma}(u)\dot{\sigma}(v)].$$

Then the neural tangent kernel for an $h$-layer network is defined as

$$\text{NTK}^{(h-1)}(\boldsymbol{x}, \boldsymbol{x}') = \sum_{i=1}^{h} \left( \Sigma^{(i-1)}(\boldsymbol{x}, \boldsymbol{x}') \cdot \prod_{k=i}^{h} \dot{\Sigma}^{(k)}(\boldsymbol{x}, \boldsymbol{x}') \right),$$

where we let $\dot{\Sigma}^{(h)}(\boldsymbol{x}, \boldsymbol{x}') = 1$ for the convenience of notations.

We compute the explicit NTK formula for a two-layer MLP with ReLU activation function by following this framework and substituting the general activation function with ReLU, i.e. $\sigma(a) = \max(0, a) = a \cdot \mathbb{I}(a \geq 0)$ and $\dot{\sigma}(a) = \mathbb{I}(a \geq 0)$.

$$\text{NTK}^{(1)}(\boldsymbol{x}, \boldsymbol{x}') = \sum_{i=1}^{2} \left( \Sigma^{(i-1)}(\boldsymbol{x}, \boldsymbol{x}') \cdot \prod_{k=i}^{h} \dot{\Sigma}^{(k)}(\boldsymbol{x}, \boldsymbol{x}') \right)$$

$$= \Sigma^{(0)}(\boldsymbol{x}, \boldsymbol{x}') \cdot \dot{\Sigma}^{(1)}(\boldsymbol{x}, \boldsymbol{x}') + \Sigma^{(1)}(\boldsymbol{x}, \boldsymbol{x}')$$

So we can get the NTK via $\Sigma^{(1)}(\boldsymbol{x}, \boldsymbol{x}')$ and $\dot{\Sigma}^{(1)}(\boldsymbol{x}, \boldsymbol{x}')$, $\Sigma^{(0)}(\boldsymbol{x}, \boldsymbol{x}')$. Precisely,

$$\Sigma^{(0)}(\boldsymbol{x}, \boldsymbol{x}') = \boldsymbol{x}^\top \boldsymbol{x}',$$

$$\wedge^{(1)}(\boldsymbol{x}, \boldsymbol{x}') = \left( \begin{array}{cc} \boldsymbol{x}^\top \boldsymbol{x} & \boldsymbol{x}^\top \boldsymbol{x}' \\ \boldsymbol{x}'^\top \boldsymbol{x} & \boldsymbol{x}'^\top \boldsymbol{x}' \end{array} \right) = \left( \begin{array}{c} \boldsymbol{x} \\ \boldsymbol{x}' \end{array} \right) \cdot \left( \begin{array}{cc} \boldsymbol{x} & \boldsymbol{x}' \end{array} \right),$$

$$\Sigma^{(1)}(\boldsymbol{x}, \boldsymbol{x}') = c \cdot \mathop{\mathbb{E}}_{u,v \sim \mathcal{N}(\boldsymbol{0}, \wedge^{(1)})} [u \cdot \mathbb{I}(u \geq 0) \cdot v \cdot \mathbb{I}(v \geq 0)].$$

To sample from $\mathcal{N}(\boldsymbol{0}, \wedge^{(1)})$, we let $L$ be a decomposition of $\wedge^{(1)}$, such that $\wedge^{(1)} = LL^\top$. Here, we can see that $L = (\boldsymbol{x}, \boldsymbol{x}')^\top$. Thus, sampling from $\mathcal{N}(\boldsymbol{0}, \wedge^{(1)})$ is equivalent to first sampling $\boldsymbol{w} \sim \mathcal{N}(\boldsymbol{0}, \boldsymbol{I})$, and output

$$L\boldsymbol{w} = \boldsymbol{w}^\top(\boldsymbol{x}, \boldsymbol{x}').$$

Then we have the equivalent sampling $(u, v) = (\boldsymbol{w}^\top \boldsymbol{x}, \boldsymbol{w}^\top \boldsymbol{x}')$. It follows that

$$\Sigma^{(1)}(\boldsymbol{x}, \boldsymbol{x}') = c \cdot \mathop{\mathbb{E}}_{\boldsymbol{w} \sim \mathcal{N}(\boldsymbol{0}, \boldsymbol{I})} [\boldsymbol{w}^\top \boldsymbol{x} \cdot \mathbb{I}(\boldsymbol{w}^\top \boldsymbol{x} \geq 0) \cdot \boldsymbol{w}^\top \boldsymbol{x}' \cdot \mathbb{I}(\boldsymbol{w}^\top \boldsymbol{x}' \geq 0)]$$

It follows from the same reasoning that

$$\dot{\Sigma}^{(1)}(\boldsymbol{x}, \boldsymbol{x}') = c \cdot \mathop{\mathbb{E}}_{\boldsymbol{w} \sim \mathcal{N}(\boldsymbol{0}, \boldsymbol{I})} \left[ \mathbb{I}\left(\boldsymbol{w}^\top \boldsymbol{x} \geq 0\right) \cdot \mathbb{I}\left(\boldsymbol{w}^\top \boldsymbol{x}' \geq 0\right) \right].$$

The neural tangent kernel for a two-layer MLP with ReLU activation is then

$$\begin{aligned}
\mathrm{NTK}^{(1)}(\boldsymbol{x}, \boldsymbol{x}') &= \Sigma^{(0)}(\boldsymbol{x}, \boldsymbol{x}') \cdot \dot{\Sigma}^{(1)}(\boldsymbol{x}, \boldsymbol{x}') + \Sigma^{(1)}(\boldsymbol{x}, \boldsymbol{x}') \\
&= c \cdot \mathop{\mathbb{E}}_{\boldsymbol{w} \sim \mathcal{N}(\boldsymbol{0}, \boldsymbol{I})} \left[ \boldsymbol{x}^\top \boldsymbol{x}' \cdot \mathbb{I}\left(\boldsymbol{w}^\top \boldsymbol{x} \geq 0\right) \cdot \mathbb{I}\left(\boldsymbol{w}^\top \boldsymbol{x}' \geq 0\right) \right] \\
&\quad + c \cdot \mathop{\mathbb{E}}_{\boldsymbol{w} \sim \mathcal{N}(\boldsymbol{0}, \boldsymbol{I})} \left[ \boldsymbol{w}^\top \boldsymbol{x} \cdot \mathbb{I}\left(\boldsymbol{w}^\top \boldsymbol{x} \geq 0\right) \cdot \boldsymbol{w}^\top \boldsymbol{x}' \cdot \mathbb{I}\left(\boldsymbol{w}^\top \boldsymbol{x}' \geq 0\right) \right].
\end{aligned}$$

Next, we use the kernel formula to compute a feature map for a two-layer MLP with ReLU activation function. Recall that by definition a valid feature map must satisfy the following condition

$$\mathrm{NTK}^{(1)}(\boldsymbol{x}, \boldsymbol{x}') = \left\langle \phi(\boldsymbol{x}), \phi(\boldsymbol{x}') \right\rangle$$

It is easy to see that the way we represent our NTK formula makes it easy to find such a decomposition. The following infinite-dimensional feature map would satisfy the requirement because the inner product of $\phi(\boldsymbol{x})$ and $\phi(\boldsymbol{x}')$ for any $\boldsymbol{x}, \boldsymbol{x}'$ would be equivalent to the expected value in NTK, after we integrate with respect to the density function of $\boldsymbol{w}$.

$$\phi(\boldsymbol{x}) = c' \left( \boldsymbol{x} \cdot \mathbb{I}\left(\boldsymbol{w}^{(k)^\top} \boldsymbol{x} \geq 0\right), \boldsymbol{w}^{(k)^\top} \boldsymbol{x} \cdot \mathbb{I}\left(\boldsymbol{w}^{(k)^\top} \boldsymbol{x} \geq 0\right), ... \right),$$

where $\boldsymbol{w}^{(k)} \sim \mathcal{N}(\boldsymbol{0}, \boldsymbol{I})$, with $k$ going to infinity. $c'$ is a constant, and $\mathbb{I}$ is the indicator function. Note that here the density of features of $\phi(\boldsymbol{x})$ is determined by the density of $\boldsymbol{w}$, i.e. Gaussian.

## C   EXPERIMENTAL DETAILS

In this section, we describe the model, data and training details for reproducing our experiments. Our experiments support all of our theoretical claims and insights.

**Overview.**   We classify our experiments into the following major categories, each of which includes several ablation studies:

1) Learning tasks where the target functions are **simple nonlinear functions** in various dimensions and training/test distributions: quadratic, cosine, square root, and l1 norm functions, with MLPs with a wide range of hyper-parameters.

   This validates our implications on MLPs generally cannot extrapolate in tasks with nonlinear target functions, unless the nonlinear function is directionally linear out-of-distribution. In the latter case, the extrapolation error is more sensitive to the hyper-parameters.

2) Computation of the **R-Squared** of MLP's learned functions along (thousands of) randomly sampled directions in out-of-distribution domain.

   This validates Theorem 1 and shows the convergence rate is very high in practice, and often happens immediately out of training range.

3) Learning tasks where the target functions are **linear functions** with MLPs. These validate Theorem 2 and Lemma 1, i.e., MLPs can extrapolate if the underlying function is linear under conditions on training distribution. This section includes four ablation studies:

   a) Training distribution satisfy the conditions in Theorem 2 and cover all directions, and hence, MLPs extrapolate.

   b) Training data distribution is **restricted** in some **directions**, e.g., restricted to be positive/negative/constant in some feature dimensions. This shows when training distribution is restrictive in directions, MLPs may fail to extrapolate.

   c) Exact extrapolation with **infinitely-wide neural networks**, i.e., exact computation with **neural tangent kernel** (NTK) on the data regime in Lemma 1. This is mainly for theoretical understanding.

4) MLPs with cosine, quadratic, and tanh activation functions.

5) Learning **maximum degree of graphs** with Graph Neural Networks. Extrapolation on graph structure, number of nodes, and node features. To show the role of architecture for extrapolation, we study the following GNN architecture regimes.

    a) GNN with graph-level max-pooling and neighbor-level sum-pooling. By Theorem 3, this GNN architecture extrapolates in max degree with appropriate training data.

    b) GNN with graph-level and neighbor-level sum-pooling. By Corollary 1, this default GNN architecture cannot extrapolate in max degree.

To show the importance of training distribution, i.e., graph structure in training set, we study the following training data regimes.

    a) Node features are **identical**, e.g., 1. In such regimes, our learning tasks only consider graph structure. We consider training sets sampled from various graph structure, and find only those satisfy conditions in Theorem 3 enables GNNs with graph-level max-pooling to extrapolate.

    b) Node features are **spurious** and continuous. This also requires extrapolation on OOD node features. GNNs with graph-level max-pooling with appropriate training sets also extrapolate to OOD spurious node features.

6) Learning the length of the **shortest path** between given source and target nodes, with Graph Neural Networks. Extrapolation on graph structure, number of nodes, and edge weights. We study the following regimes.

    a) Continuous features. Edge and node features are real values. This regime requires extrapolating to graphs with edge weights out of training range.

Test graphs are all sampled from the "general graphs" family with a diverse range of structure. Regarding the type of training graph structure, we consider two schemes. Both schemes show a U-shape curve of extrapolation error with respect to the sparsity of training graphs.

    a) Specific graph structure: path, cycle, tree, expander, ladder, complete graphs, general graphs, 4-regular graphs.

    b) Random graphs with a range of probability $p$ of an edge between any two nodes. Smaller $p$ samples sparse graphs and large $p$ samples dense graphs.

7) **Physical reasoning** of the $n$-**Body problem** in the orbit setting with Graph Neural Networks. We show that GNNs on the original features from previous works fail to extrapolate to unseen masses and distances. On the other hand, we show extrapolation can be achieved via an improved representation of the input edge features. We consider the following extrapolation regimes.

    a) Extrapolation on the masses of the objects.

    b) Extrapolation on the distances between objects.

We consider the following two *input representation* schemes to compare the effects of how representation helps extrapolation.

    a) Original features. Following previous works on solving $n$-body problem with GNNs, the edge features are simply set to $0$.

    b) Improved features. We show although our edge features do not bring in new information, it helps extrapolation.

## C.1 Learning Simple Non-Linear Functions

**Dataset details.** We consider four tasks where the underlying functions are simple non-linear functions $g : \mathbb{R}^d \to \mathbb{R}$. Given an input $\boldsymbol{x} \in \mathbb{R}^d$, the label is computed by $y = g(\boldsymbol{x})$ for all $\boldsymbol{x}$. We consider the following four families of simple functions $g$.

    a) Quadratic functions $g(\boldsymbol{x}) = \boldsymbol{x}^\top A \boldsymbol{x}$. In each dataset, we randomly sample $A$. In the simplest case where $A = I$, $g(\boldsymbol{x}) = \sum_{i=1}^d x_i^2$.

    a) Cosine functions $g(\boldsymbol{x}) = \sum_{i=1}^d \cos\left(2\pi \cdot \boldsymbol{x}_i\right)$.

    c) Square root functions $g(\boldsymbol{x}) = \sum_{i=1}^d \sqrt{\boldsymbol{x}_i}$. Here, the domain $\mathcal{X}$ of $\boldsymbol{x}$ is restricted to the space in $\mathbb{R}^d$ with non-negative value in each dimension.

    d) L1 norm functions $g(\boldsymbol{x}) = |\boldsymbol{x}|_1 = \sum_{i=1}^{d} |\boldsymbol{x}_i|$.

We sample each dataset of a task by considering the following parameters

    a) The shape and support of training, validation, and test data distributions.

        i) Training, validation, and test data are uniformly sampled from a hyper-cube. Training and validation data are sampled from $[-a, a]^d$ with $a \in \{0.5, 1.0\}$, i.e., each dimension of $\boldsymbol{x} \in \mathbb{R}^d$ is uniformly sampled from $[-a, a]$. Test data are sampled from $[-a, a]^d$ with $a \in \{2.0, 5.0, 10.0\}$.

        ii) Training and validation data are uniformly sampled from a sphere, where every point has $L2$ distance $r$ from the origin. We sample $r$ from $r \in \{0.5, 1.0\}$. Then, we sample a random Gaussian vector $\boldsymbol{q}$ in $\mathbb{R}^d$. We obtain the training or validation data $\boldsymbol{x} = \boldsymbol{q}/\|\boldsymbol{q}\|_2 \cdot r$. This corresponds to uniform sampling from the sphere.

        Test data are sampled (non-uniformly) from a hyper-ball. We first sample $r$ uniformly from $[0.0, 2.0]$, $[0.0, 5.0]$, and $[0.0, 10.0]$. Then, we sample a random Gaussian vector $\boldsymbol{q}$ in $\mathbb{R}^d$. We obtain the test data $\boldsymbol{x} = \boldsymbol{q}/\|\boldsymbol{q}\|_2 \cdot r$. This corresponds to (non-uniform) sampling from a hyper-ball in $\mathbb{R}^d$.

    b) We sample $20,000$ training data, $1,000$ validation data, and $20,000$ test data.

    c) We sample input dimension $d$ from $\{1, 2, 8\}$.

    d) For quadratic functions, we sample the entries of $A$ uniformly from $[-1, 1]$.

**Model and hyperparameter settings.** We consider the multi-layer perceptron (MLP) architecture.

$$\text{MLP}(\boldsymbol{x}) = \boldsymbol{W}^{(d)} \cdot \sigma \left( \boldsymbol{W}^{(d-1)} \sigma \left( ...\sigma \left( \boldsymbol{W}^{(1)} \boldsymbol{x} \right) \right) \right)$$

We search the following hyper-parameters for MLPs

    a) Number of layers $d$ from $\{2, 4\}$.

    b) Width of each $\boldsymbol{W}^{(k)}$ from $\{64, 128, 512\}$.

    c) Initialization schemes.

        i) The default initialization in PyTorch.

        ii) The initialization scheme in neural tangent kernel theory, i.e., we sample entries of $\boldsymbol{W}^k$ from $\mathcal{N}(0, 1)$ and scale the output after each $\boldsymbol{W}^{(k)}$ by $\sqrt{\frac{2}{d_k}}$, where $d_k$ is the output dimension of $\boldsymbol{W}^{(k)}$.

    d) Activation function $\sigma$ is set to ReLU.

We train the MLP with the mean squared error (MSE) loss, and Adam and SGD optimizer. We consider the following hyper-parameters for training

    a) Initial learning rate from $\{5e-2, 1e-2, 5e-3, 1e-3\}$. Learning rate decays $0.5$ for every $50$ epochs

    b) Batch size from $\{32, 64, 128\}$.

    c) Weight decay is set to $1e-5$.

    d) Number of epochs is set to $250$.

**Test error and model selection.** For each dataset and architecture, training hyper-parameter setting, we perform model selection via validation set, i.e., we report the test error by selecting the epoch where the model achieves the best validation error. Note that our validation sets always have the same distribution as the training sets.

We train our models with the MSE loss. Because we sample test data from different ranges, the mean absolute percentage error (MAPE) loss, which scales the error by the actual value, better measures the extrapolation performance

$$\text{MAPE} = \frac{1}{n} \left| \frac{A_i - F_i}{A_i} \right|,$$

where $A_i$ is the actual value and $F_i$ is the predicted value. Hence, in our experiments, we also report the MAPE.

## C.2 R-squared for Out-of-distribution Directions

We perform linear regression to fit the predictions of MLPs along randomly sampled directions in out-of-distribution regions, and compute the R-squared (or $R^2$) for these directions. This experiment is to validate Theorem 1 and show that the convergence rate (to a linear function) is very high in practice.

**Definition.** R-squared, also known as coefficient of determination, assesses how strong the linear relationship is between input and output variables. The closer R-squared is to 1, the stronger the linear relationship is, with 1 being perfectly linear.

**Datasets and models.** We perform the R-squared computation on over $2,000$ combinations of datasets, test/train distributions, and hyper-parameters, e.g., learning rate, batch size, MLP layer, width, initialization. These are described in Appendix C.1.

**Computation.** For each combination of dataset and model hyper-parameters as described in Section C.1, we save the trained MLP model $f : \mathbb{R}^d \to \mathbb{R}$. For each dataset and model combination, we then randomly sample $5,000$ directions via Gaussian vectors $\mathcal{N}(\mathbf{0}, \boldsymbol{I})$. For each of these directions $\boldsymbol{w}$, we compute the intersection point $\boldsymbol{x_w}$ of direction $\boldsymbol{w}$ and the training data distribution support (specified by a hyper-sphere or hyper-cube; see Section C.1 for details).

We then collect 100 predictions of the trained MLP $f$ along direction $\boldsymbol{w}$ (assume $\boldsymbol{w}$ is normalized) with

$$\left\{ \left( \boldsymbol{x_w} + k \cdot \frac{r}{10} \cdot \boldsymbol{w} \right), f \left( \boldsymbol{x_w} + k \cdot \frac{r}{10} \cdot \boldsymbol{w} \right) \right\}_{k=0}^{100}, \tag{75}$$

where $r$ is the range of training data distribution support (see Section C.1). We perform linear regression on these predictions in Eqn. 75, and obtain the R-squared.

**Results.** We obtain the R-squared for each combination of dataset, model and training setting, and randomly sampled direction. For the tasks of learning the simple non-linear functions, we confirm that more than $96\%$ of the R-squared results are above $0.99$. This empirically confirms Theorem 1 and shows that the convergence rate is in fact fast in practice. Along most directions, MLP's learned function becomes linear immediately out of the training data support.

## C.3 Learning Linear Functions

**Dataset details.** We consider the tasks where the underlying functions are linear $g : \mathbb{R}^d \to \mathbb{R}$. Given an input $\boldsymbol{x} \in \mathbb{R}^d$, the label is computed by $y = g(\boldsymbol{x}) = A\boldsymbol{x}$ for all $\boldsymbol{x}$. For each dataset, we sample the following parameters

    a) We sample $10,000$ training data, $1,000$ validation data, and $2,000$ test data.

    b) We sample input dimension $d$ from $\{1, 2, 32\}$.

    c) We sample entries of $A$ uniformly from $[-a, a]$, where we sample $a \in \{5.0, 10.0\}$.

    d) The shape and support of training, validation, and test data distributions.

        i) Training, validation, and test data are uniformly sampled from a hyper-cube. Training and validation data are sampled from $[-a, a]^d$ with $a \in \{5.0, 10.0\}$, i.e., each dimension of $\boldsymbol{x} \in \mathbb{R}^d$ is uniformly sampled from $[-a, a]$. Test data are sampled from $[-a, a]^d$ with $a \in \{20.0, 50.0\}$.

        ii) Training and validation data are uniformly sampled from a sphere, where every point has $L2$ distance $r$ from the origin. We sample $r$ from $r \in \{5.0, 10.0\}$. Then, we sample a random Gaussian vector $\boldsymbol{q}$ in $\mathbb{R}^d$. We obtain the training or validation data $\boldsymbol{x} = \boldsymbol{q}/\|\boldsymbol{q}\|_2 \cdot r$. This corresponds to uniform sampling from the sphere.

Test data are sampled (non-uniformly) from a hyper-ball. We first sample $r$ uniformly from $[0.0, 20.0]$ and $[0.0, 50.0]$,. Then, we sample a random Gaussian vector $\boldsymbol{q}$ in $\mathbb{R}^d$. We obtain the test data $\boldsymbol{x} = \boldsymbol{q}/\|\boldsymbol{q}\|_2 \cdot r$. This corresponds to (non-uniform) sampling from a hyper-ball in $\mathbb{R}^d$.

e) We perform ablation study on how the training distribution support misses directions. The test distributions remain the same as in d).

   i) We restrict the first dimension of any training data $\boldsymbol{x}_i$ to a fixed number $0.1$, and randomly sample the remaining dimensions according to d).

   ii) We restrict the first $k$ dimensions of any training data $\boldsymbol{x}_i$ to be positive. For input dimension 32, we only consider the hyper-cube training distribution, where we sample the first $k$ dimensions from $[0, a]$ and sample the remaining dimensions from $[-a, a]$. For input dimensions 1 and 2, we consider both hyper-cube and hyper-sphere training distribution by performing rejection sampling. For input dimension 2, we consider $k$ from $\{1, 2\}$. For input dimension 32, we consider $k$ from $\{1, 16, 32\}$.

   iii) We restrict the first $k$ dimensions of any training data $\boldsymbol{x}_i$ to be negative. For input dimension 32, we only consider the hyper-cube training distribution, where we sample the first $k$ dimensions from $[-a, 0]$ and sample the remaining dimensions from $[-a, a]$. For input dimensions 1 and 2, we consider both hyper-cube and hyper-sphere training distribution by performing rejection sampling. For input dimension 2, we consider $k$ from $\{1, 2\}$. For input dimension 32, we consider $k$ from $\{1, 16, 32\}$.

**Model and hyperparameter settings.** For the regression task, we search the same set of hyper-parameters as those in simple non-linear functions (Section C.1).We report the test error with the same validation procedure as in Section C.1.

**Exact computation with neural tangent kernel** Our experiments with MLPs validate Theorem 2 asymptotic extrapolation for neural networks trained in regular regimes. Here, we also validate Lemma 1, exact extrapolation with finite data regime, by training an infinitely-wide neural network. That is, we directly perform the kernel regression with the neural tangent kernel (NTK). This experiment is mainly of theoretical interest.

We sample the same test set as in our experiments with MLPs. For training set, we sample $2d$ training examples according to the conditions in Lemma 1. Specifically, we first sample an orthogonal basis and their opposite vectors $\boldsymbol{X} = \{\boldsymbol{e}_i, -\boldsymbol{e}_i\}_{i=1}^d$. We then randomly sample 100 orthogonal transform matrices $Q$ via the QR decomposition. Our training samples are $Q\boldsymbol{X}$, i.e., multiply each point in $\boldsymbol{X}$ by $Q$. This gives 100 training sets with $2d$ data points satisfying the condition in Lemma 1.

We perform kernel regression on these training sets using a two-layer neural tangent kernel (NTK). Our code for exact computation of NTK is adapted from Arora et al. (2020); Novak et al. (2020). We verify that the test losses are all precisely 0, up to machine precision. This empirically confirms Lemma 1.

Note that due to the difference of hyper-parameter settings in different implementations of NTK, to reproduce our experiments and achieve zero test error, the implementation by Arora et al. (2020) is assumed.

### C.4 MLPs with cosine, quadratic, and tanh Activation

This section describes the experimental settings for extrapolation experiments for MLPs with cosine, quadratic, and tanh activation functions. We train MLPs to learn the following functions:

a) Quadratic function $g(\boldsymbol{x}) = \boldsymbol{x}^\top A \boldsymbol{x}$, where $A$ is a randomly sampled matrix.

b) Cosine function $g(\boldsymbol{x}) = \sum_{i=1}^d \cos(2\pi \cdot \boldsymbol{x}_i)$.

c) Hyperbolic tangent function $g(\boldsymbol{x}) = \sum_{i=1}^d \tanh(\boldsymbol{x}_i)$.

d) Linear function $g(\boldsymbol{x}) = W\boldsymbol{x} + b$.

**Dataset details.** We use 20,000 training, 1,000 validation, and 20,000 test data. For quadratic, we sample input dimension $d$ from $\{1, 8\}$, training and validation data from $[-1, 1]^d$, and test data from $[-5, 5]^d$. For cosine, we sample input dimension $d$ from $\{1, 2\}$, training and validation data from $[-100, 100]^d$, and test data from $[-200, 200]^d$. For tanh, we sample input dimension $d$ from $\{1, 8\}$, training and validation data from $[-100, 100]^d$, and test data from $[-200, 200]^d$. For linear, we use a subset of datasets from Appendix C.3: 1 and 8 input dimensions with hyper-cube training distributions.

**Model and hyperparameter settings.** We use the same hyperparameters from Appendix C.1, except we fix the batch size to 128, as the batch size has minimal impact on models. MLPs with cos activation is hard to optimize, so we only report models with training MAPE less than 1.

## C.5 MAX DEGREE

**Dataset details.** We consider the task of finding the maximum degree on a graph. Given any input graph $G = (V, E)$, the label is computed by the underlying function $y = g(G) = \max_{u \in G} \sum_{v \in \mathcal{N}(u)} 1$. For each dataset, we sample the graphs and node features with the following parameters

a) Graph structure for training and validation sets. For each dataset, we consider one of the following graph structure: path graphs, cycles, ladder graphs, 4-regular random graphs, complete graphs, random trees, expanders (here we use random graphs with $p = 0.8$ as they are expanders with high probability), and general graphs (random graphs with $p = 0.1$ to 0.9 with equal probability for a broad range of graph structure). We use the networkx library for sampling graphs.

b) Graph structure for test set. We consider the general graphs (random graphs with $p = 0.1$ to 0.9 with equal probability).

c) The number of vertices of graphs $|V|$ for training and validation sets are sampled uniformly from $[20...30]$. The number of vertices of graphs $|V|$ for test set is sampled uniformly from $[50..100]$.

d) We consider two schemes for node features.

   i) Identical features. All nodes in training, validation and set sets have uniform feature 1.

   ii) Spurious (continuous) features. Node features in training and validation sets are sampled uniformly from $[-5.0, 5.0]^3$, i.e., a three-dimensional vector where each dimension is sampled from $[-5.0, 5.0]$. There are two schemes for test sets, in the first case we do not extrapolate node features, so we sample node features uniformly from $[-5.0, 5.0]^3$. In the second case we extrapolate node features, we sample node features uniformly from $[-10.0, 10.0]^3$.

e) We sample $5,000$ graphs for training, $1,000$ graphs for validation, and $2,500$ graphs for testing.

**Model and hyperparameter settings.** We consider the following Graph Neural Network (GNN) architecture. Given an input graph $G$, GNN learns the output $h_G$ by first iteratively aggregating and transforming the neighbors of all node vectors $h_u^{(k)}$ (vector for node $u$ in layer $k$), and perform a max or sum-pooling over all node features $h_u$ to obtain $h_G$. Formally, we have

$$h_u^{(k)} = \sum_{v \in \mathcal{N}(u)} \text{MLP}^{(k)} \left( h_v^{(k-1)}, h_u^{(k-1)} \right), \quad h_G = \text{MLP}^{(K+1)} \left( \text{graph-pooling}\{h_u^{(K)} : u \in G\} \right).$$

(76)

Here, $\mathcal{N}(u)$ denotes the neighbors of $u$, $K$ is the number of GNN iterations, and graph-pooling is a hyper-parameter with choices as max or sum. $h_u^{(0)}$ is the input node feature of node $u$. We search the following hyper-parameters for GNNs

a) Number of GNN iterations $K$ is 1.

b) Graph pooling is from max or sum.

c) Width of all MLPs are set to 256.

    d) The number of layers for $\text{MLP}^{(k)}$ with $k = 1..K$ are set to 2. The number of layers for $\text{MLP}^{(K+1)}$ is set to 1.

We train the GNNs with the mean squared error (MSE) loss, and Adam and SGD optimizer. We search the following hyper-parameters for training

    a) Initial learning rate is set to $0.01$.

    b) Batch size is set to $64$.

    c) Weight decay is set to $1e-5$.

    d) Number of epochs is set to $300$ for graphs with continuous node features, and $100$ for graphs with uniform node features.

**Test error and model selection.** For each dataset and architecture, training hyper-parameter setting, we perform model selection via validation set, i.e., we report the test error by selecting the epoch where the model achieves the best validation error. Note that our validation sets always have the same distribution as the training sets. Again, we report the MAPE for test error as in MLPs.

## C.6 SHORTEST PATH

**Dataset details.** We consider the task of finding the length of the shortest path on a graph, from a given source to target nodes. Given any graph $G = (V, E)$, the node features, besides regular node features, encode whether a node is source $s$, and whether a node is target $t$. The edge features are a scalar representing the edge weight. For unweighted graphs, all edge weights are $1$. Then the label $y = g(G)$ is the length of the shortest path from $s$ to $t$ on $G$.

For each dataset, we sample the graphs and node, edge features with the following parameters

    a) Graph structure for training and validation sets. For each dataset, we consider one of the following graph structure: path graphs, cycles, ladder graphs, 4-regular random graphs, complete graphs, random trees, expanders (here we use random graphs with $p = 0.6$ which are expanders with high probability), and general graphs (random graphs with $p = 0.1$ to $0.9$ with equal probability for a broad range of graph structure). We use the networkx library for sampling graphs.

    b) Graph structure for test set. We consider the general graphs (random graphs with $p = 0.1$ to $0.9$ with equal probability).

    c) The number of vertices of graphs $|V|$ for training and validation sets are sampled uniformly from $[20...40]$. The number of vertices of graphs $|V|$ for test set is sampled uniformly from $[50..70]$.

    d) We consider the following scheme for node and edge features. All edges have continuous weights. Edge weights for training and validation graphs are sampled from $[1.0, 5.0]$. There are two schemes for test sets, in the first case we do not extrapolate edge weights, so we sample edge weights uniformly from $[1.0, 5.0]$. In the second case we extrapolate edge weights, we sample edge weights uniformly from $[1.0, 10.0]$. All node features are $[h, \mathbb{I}(v = s), \mathbb{I}(v = t)]$ with $h$ sampled from $[-5.0, 5.0]$.

    e) After sampling a graph and edge weights, we sample source $s$ and $t$ by randomly sampling $s, t$ and selecting the first pair $s, s$ whose shortest path involves at most 3 hops. This enables us to solve the task using GNNs with 3 iterations.

    f) We sample $10,000$ graphs for training, $1,000$ graphs for validation, and $2,500$ graphs for testing.

We also consider the ablation study of training on random graphs with different $p$. We consider $p = 0.05..1.0$ and report the test error curve. The other parameters are the same as described above.

**Model and hyperparameter settings.** We consider the following Graph Neural Network (GNN) architecture. Given an input graph $G$, GNN learns the output $h_G$ by first iteratively aggregating and

transforming the neighbors of all node vectors $h_u^{(k)}$ (vector for node $u$ in layer $k$), and perform a max or sum-pooling over all node features $h_u$ to obtain $h_G$. Formally, we have

$$h_u^{(k)} = \min_{v \in \mathcal{N}(u)} \text{MLP}^{(k)} \left( h_v^{(k-1)}, h_u^{(k-1)}, w_{(u,v)} \right), \quad h_G = \text{MLP}^{(K+1)} \left( \min_{u \in G} h_u \right). \quad (77)$$

Here, $\mathcal{N}(u)$ denotes the neighbors of $u$, $K$ is the number of GNN iterations, and for neighbor aggregation we run both min and sum. $h_u^{(0)}$ is the input node feature of node $u$. $w_{(u,v)}$ is the input edge feature of edge $(u, v)$. We search the following hyper-parameters for GNNs

a) Number of GNN iterations $K$ is set to 3.

b) Graph pooling is set to min.

c) Neighobr aggregation is selected from min and sum.

d) Width of all MLPs are set to 256.

e) The number of layers for $\text{MLP}^{(k)}$ with $k = 1..K$ are set to 2. The number of layers for $\text{MLP}^{(K+1)}$ is set to 1.

We train the GNNs with the mean squared error (MSE) loss, and Adam and SGD optimizer. We consider the following hyper-parameters for training

a) Initial learning rate is set to $0.01$.

b) Batch size is set to $64$.

c) Weight decay is set to $1e - 5$.

d) Number of epochs is set to $250$.

We perform the same model selection and validation as in Section C.5.

## C.7   N-BODY PROBLEM

**Task description.** The n-body problem asks a neural network to predict how n stars in a physical system evolves according to physics laws. That is, we train neural networks to predict properties of future states of each star in terms of next frames, e.g., $0.001$ seconds.

Mathematically, in an n-body system $S = \{X_i\}_{i=1}^n$, such as solar systems, all n stars $\{X_i\}_{i=1}^n$ exert distance and mass-dependent gravitational forces on each other, so there were $n(n-1)$ relations or forces in the system. Suppose $X_i$ at time $t$ is at position $\boldsymbol{x}_i^t$ and has velocity $\boldsymbol{v}_i^t$. The overall forces a star $X_i$ receives from other stars is determined by physics laws as the following

$$\boldsymbol{F}_i^t = G \cdot \sum_{j \neq i} \frac{m_i \times m_j}{\|\boldsymbol{x}_i^t - \boldsymbol{x}_j^t\|_2^3} \cdot \left( \boldsymbol{x}_j^t - \boldsymbol{x}_i^t \right), \quad (78)$$

where $G$ is the gravitational constant, and $m_i$ is the mass of star $X_i$. Then acceralation $\boldsymbol{a}_i^t$ is determined by the net force $\boldsymbol{F}_i^t$ and the mass of star $m_i$

$$\boldsymbol{a}_i^t = \boldsymbol{F}_i^t / m_i \quad (79)$$

Suppose the velocity of star $X_i$ at time $t$ is $\boldsymbol{v}_i^t$. Then assuming the time steps $dt$, i.e., difference between time frames, are sufficiently small, the velocity at the next time frame $t+1$ can be approximated by

$$\boldsymbol{v}_i^{t+1} = \boldsymbol{v}_i^t + \boldsymbol{a}_i^t \cdot dt. \quad (80)$$

Given $m_i$, $\boldsymbol{x}_i^t$, and $\boldsymbol{v}_i^t$, our task asks the neural network to predict $\boldsymbol{v}_i^{t+1}$ for all stars $X_i$. In our task, we consider two extrapolation schemes

a) The distances between stars $\|\boldsymbol{x}_i^t - \boldsymbol{x}_j^t\|_2$ are out-of-distribution for test set, i.e., different sampling ranges from the training set.

b) The masses of stars $m_i$ are out-of-distribution for test set, i.e., different sampling ranges from the training set.

Here, we use a physics engine that we code in Python to simulate and sample the inputs and labels. We describe the dataset details next.

**Dataset details.** We first describe the simulation and sampling of our training set. We sample $100$ videos of n-body system evolution, each with $500$ rollout, i.e., time steps. We consider the orbit situation: there exists a huge center star and several other stars. We sample the initial states, i.e., position, velocity, masses, acceleration etc according to the following parameters.

    a) The mass of the center star is $100kg$.

    b) The masses of other stars are sampled from $[0.02, 9.0]kg$.

    c) The number of stars is $3$.

    d) The initial position of the center star is $(0.0, 0.0)$.

    d) The initial positions $\boldsymbol{x}_i^t$ of other objects are randomly sampled from all angles, with a distance in $[10.0, 100.0]m$.

    e) The velocity of the center star is $\boldsymbol{0}$.

    f) The velocities of other stars are perpendicular to the gravitational force between the center star and itself. The scale is precisely determined by physics laws to ensure the initial state is an orbit system.

For each video, after we get the initial states, we continue to rollout the next frames according the physics engine described above. We perform rejection sampling of the frames to ensure that all pairwise distances of stars in a frame are at least $30m$. We guarantee that there are $10,000$ data points in the training set.

The validation set has the same sampling and simultation parameters as the training set. We have $2,500$ data points in the validation set.

For test set, we consider two datasets, where we respectively have OOD distances and masses. We have $5,000$ data points for each dataset.

    a) We sample the distance OOD test set to ensure all pairwise distances of stars in a frame are from $[1..20]m$, but have in-distribution masses.

    b) We sample the mass OOD test set as follows

        i) The mass of the center star is $200kg$, i.e., twice of that in the training set.

        ii) The masses of other stars are sampled from $[0.04, 18.0]kg$, compared to $[0.02, 9.0]kg$ in the training set.

        iii) The distances are in-distribution, i.e., same sampling process as training set.

**Model and hyperparameter settings.** We consider the following one-iteration Graph Neural Network (GNN) architecture, a.k.a. Interaction Networks. Given a collection of stars $S = \{X_i\}_{i=1}^n$, our GNN runs on a complete graph with nodes being the stars $X_i$. GNN learns the star (node) representations by aggregating and transforming the interactions (forces) of all other node vectors

$$o_u = \text{MLP}^{(2)} \left( \sum_{v \in S \setminus \{u\}} \text{MLP}^{(1)} \left( h_v, h_u, w_{(u,v)} \right) \right). \tag{81}$$

Here, $h_v$ is the input feature of node $v$, including mass, position and velocity

$$h_v = (m_v, \boldsymbol{x}_v, \boldsymbol{v}_v)$$

$w_{(u,v)}$ is the input edge feature of edge $(u, v)$. The loss is computed and backpropagated via the MSE loss of

$$\|[o_1, ..., o_n] - [ans_1, .., ans_n]\|_2,$$

where $o_i$ denotes the output of GNN for node $i$, and $ans_i$ denotes the true label for node $i$ in the next frame.

We search the following hyper-parameters for GNNs

    a) Number of GNN iterations is set to $1$.

b) Width of all MLPs are set to 128.

c) The number of layers for $\text{MLP}^{(1)}$ is set to 4. The number of layers for $\text{MLP}^{(2)}$ is set to 2.

d) We consider *two representations* of edge/relations $w_{(i,j)}$.

    i) The first one is simply 0.

    ii) The better representation, which makes the underlying target function more linear, is

$$w_{(i,j)} = \frac{m_j}{\|\boldsymbol{x}_i^t - \boldsymbol{x}_j^t\|_2^3} \cdot \left(\boldsymbol{x}_j^t - \boldsymbol{x}_i^t\right)$$

We train the GNN with the mean squared error (MSE) loss, and Adam optimizer. We search the following hyper-parameters for training

a) Initial learning rate is set to $0.005$. learning rate decays $0.5$ for every 50 epochs

b) Batch size is set to 32.

c) Weight decay is set to $1e-5$.

d) Number of epochs is set to $2,000$.

# D    VISUALIZATION AND ADDITIONAL EXPERIMENTAL RESULTS

## D.1    VISUALIZATION RESULTS

In this section, we show additional visualization results of the MLP's learned function out of training distribution (in **black color**) v.s. the underlying true function (in **grey color**). We color the predictions in training distribution in **blue color**.

In general, MLP's learned functions agree with the underlying true functions in training range (blue). This is explained by in-distribution generalization arguments. When out of distribution, the MLP's learned functions become linear along directions from the origin. We explain this OOD directional linearity behavior in Theorem 1.

Finally, we show additional experimental results for graph-based reasoning tasks.

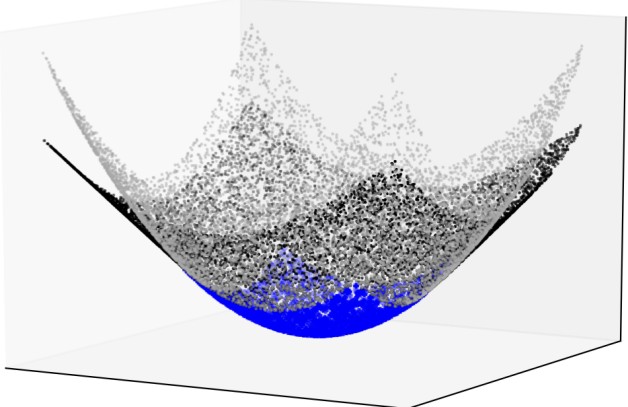

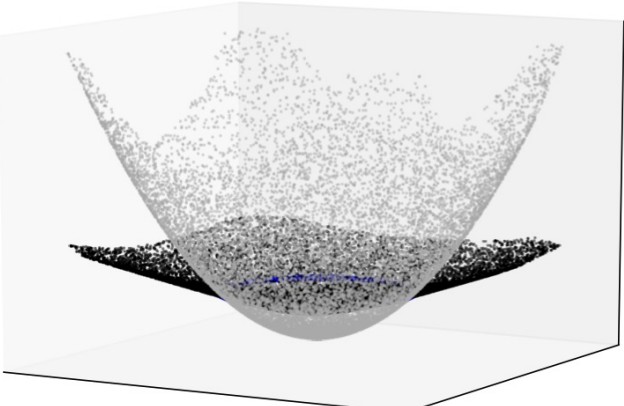

Figure 8: **(Quadratic function).** Both panels show the learned v.s. true $y = x_1^2 + x_2^2$. In each figure, we color OOD predictions by MLPs in black, underlying function in grey, and in-distribution predictions in blue. The support of training distribution is a square (cube) for the top panel, and is a circle (sphere) for the bottom panel.

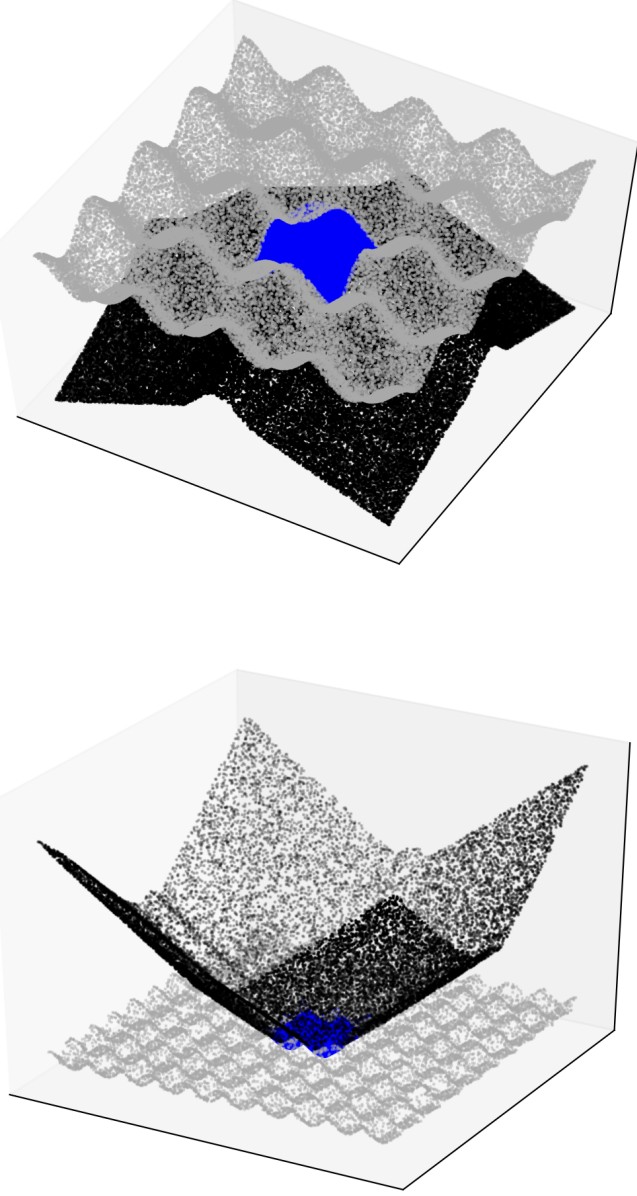

Figure 9: **(Cos function).** Both panels show the learned v.s. true $y = \cos(2\pi \cdot x_1) + \cos(2\pi \cdot x_2)$. In each figure, we color OOD predictions by MLPs in black, underlying function in grey, and in-distribution predictions in blue. The support of training distribution is a square (cube) for both top and bottom panels, but with different ranges.

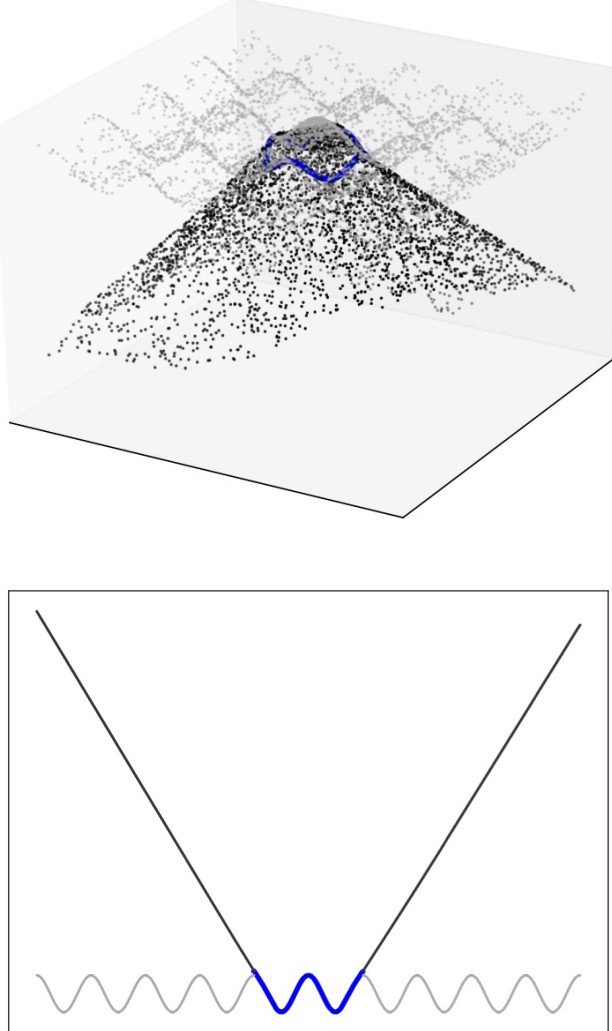

Figure 10: **(Cos function).** Top panel shows the learned v.s. true $y = \cos(2\pi \cdot x_1) + \cos(2\pi \cdot x_2)$ where the support of training distribution is a circle (sphere). Bottom panel shows results for cosine in 1D, i.e. $y = \cos(2\pi \cdot x)$. In each figure, we color OOD predictions by MLPs in black, underlying function in grey, and in-distribution predictions in blue.

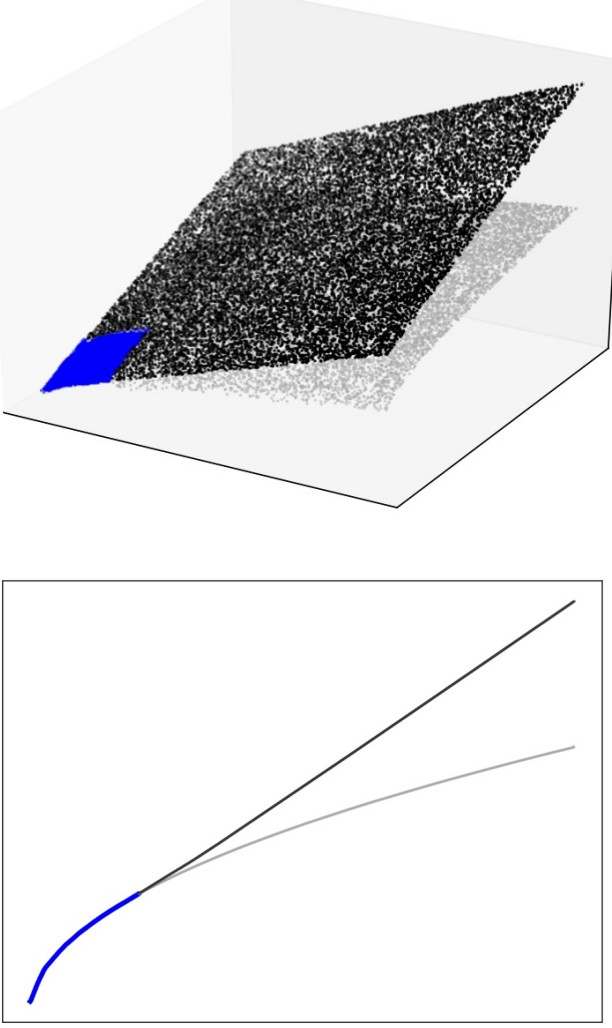

Figure 11: **(Sqrt function).** Top panel shows the learned v.s. true $y = \sqrt{x_1} + \sqrt{x_2}$ where the support of training distribution is a square (cube). Bottom panel shows the results for the square root function in 1D, i.e. $y = \sqrt{x}$. In each figure, we color OOD predictions by MLPs in black, underlying function in grey, and in-distribution predictions in blue.

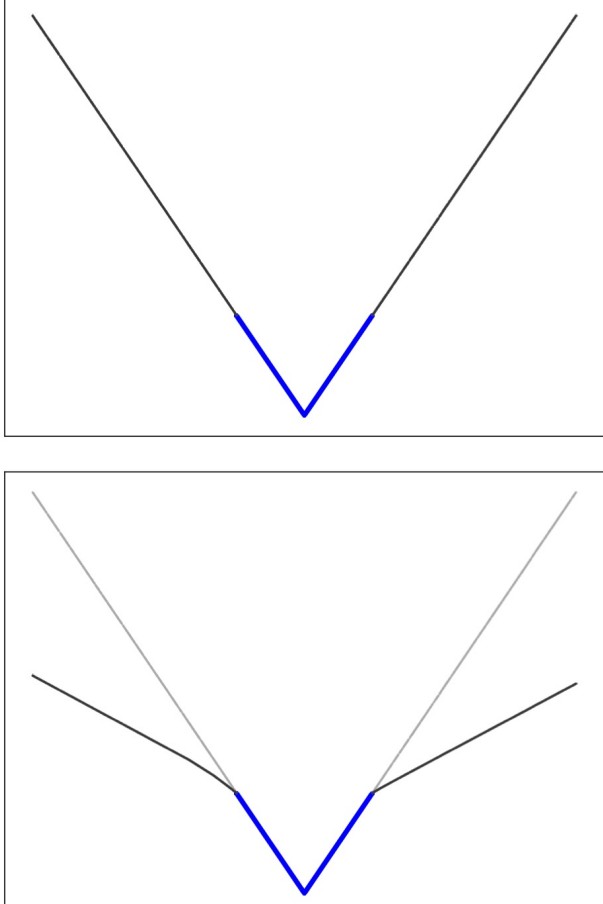

Figure 12: **(L1 function).** Both panels show the learned v.s. true $y = |x|$. In the top panel, the MLP successfully learns to extrapolate the absolute function. In the bottom panel, an MLP with different hyper-parameters fails to extrapolate. In each figure, we color OOD predictions by MLPs in black, underlying function in grey, and in-distribution predictions in blue.

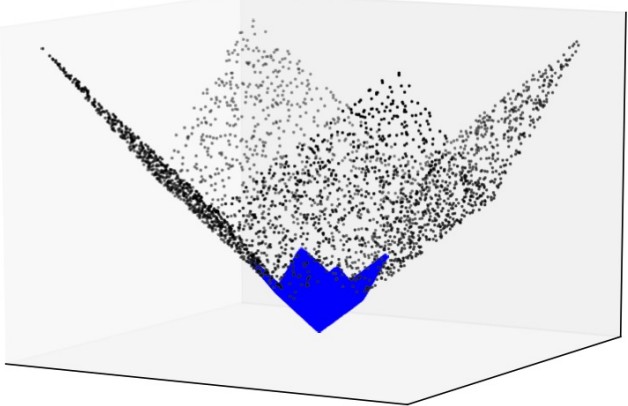

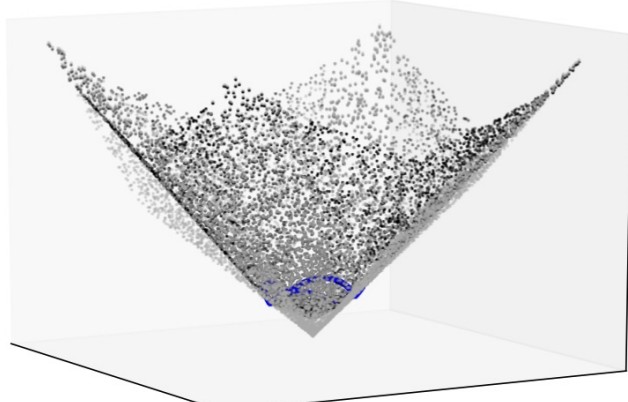

Figure 13: **(L1 function).** Both panels show the learned v.s. true $y = |x_1| + |x_2|$. In the top panel, the MLP successfully learns to extrapolate the l1 norm function. In the bottom panel, an MLP with different hyper-parameters fails to extrapolate. In each figure, we color OOD predictions by MLPs in black, underlying function in grey, and in-distribution predictions in blue.

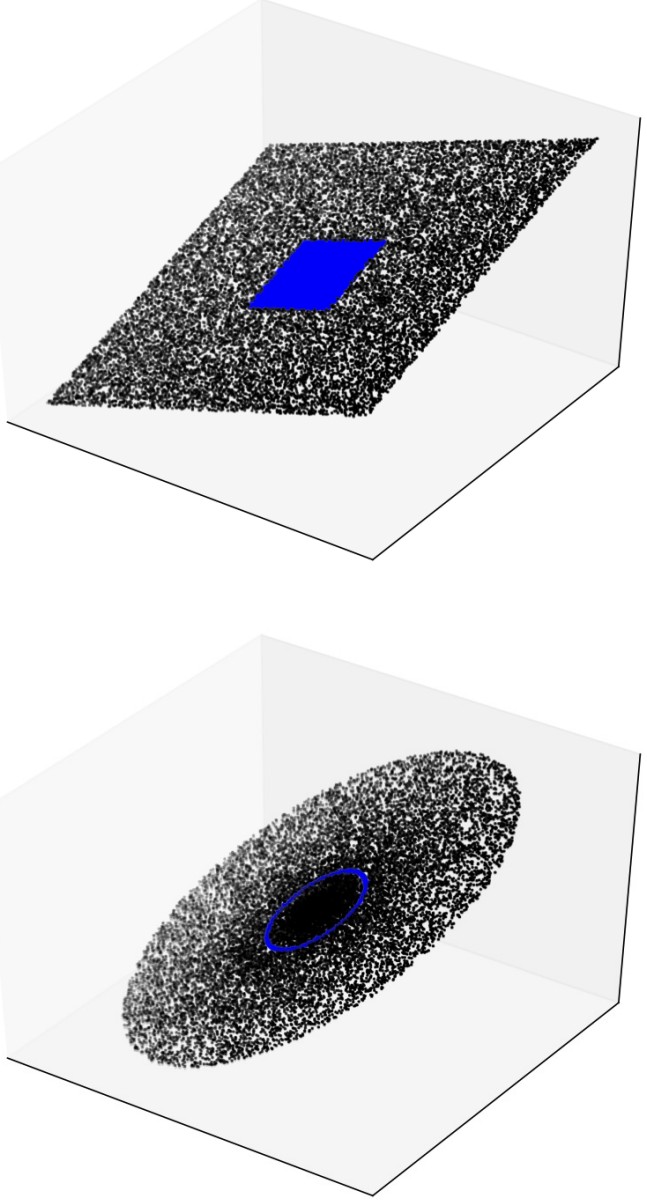

Figure 14: (**Linear function**). Both panels show the learned v.s. true $y = x_1 + x_2$, with the support of training distributions being square (cube) for top panel, and circle (sphere) for bottom panel. MLPs successfully extrapolate the linear function with both training distributions. This is explained by Theorem 2: both sphere and cube intersect all directions. In each figure, we color OOD predictions by MLPs in black, underlying function in grey, and in-distribution predictions in blue.

## D.2 EXTRA EXPERIMENTAL RESULTS

In this section, we show additional experimental results.

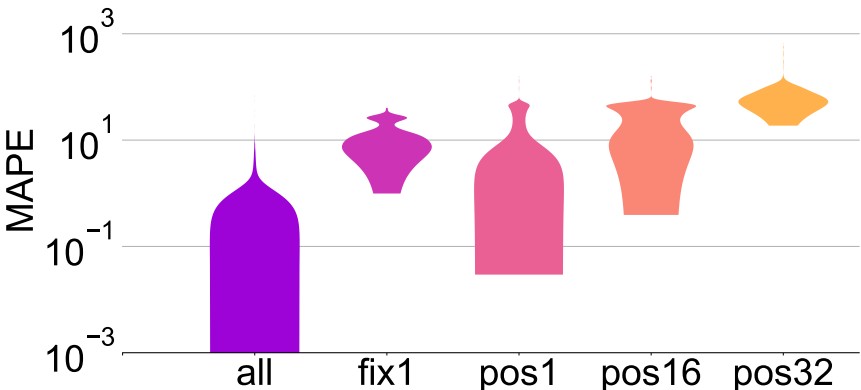

Figure 15: **Density plot of the test errors in MAPE.** The underlying functions are linear, but we train MLPs on different distributions, whose support potentially miss some directions. The training support for "all" are hyper-cubes that intersect all directions. In "fix1", we set the first dimension of training data to a fixed number. In "posX", we restrict the first X dimensions of training data to be positive. We can see that MLPs trained on "all" extrapolate the underlying linear functions, but MLPs trained on datasets with missing directions, i.e., "fix1" and "posX", often cannot extrapolate well.

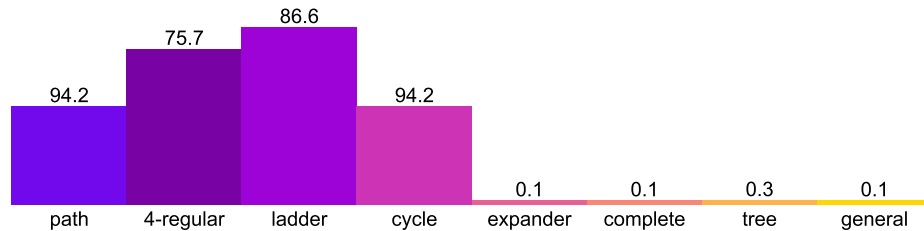

Figure 16: **Maximum degree: continuous and "spurious" node features.** Here, each node has a node feature in $\mathbb{R}^3$ that shall not contribute to the answer of maximum degree. GNNs with graph-level max-pooling extrapolate to graphs with OOD node features and graph structure, graph sizes, if trained on graphs that satisfy the condition in Theorem 3.

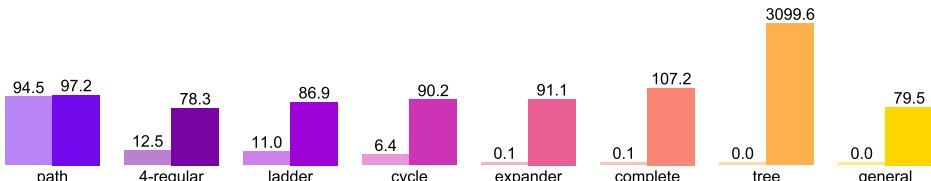

Figure 17: **Maximum degree: max-pooling v.s. sum-pooling.** In each sub-figure, left column shows test errors for GNNs with graph-level max-pooling; right column shows test errors for GNNs with graph-level sum-pooling. x-axis shows the graph structure covered in training set. GNNs with sum-pooling fail to extrapolate, validating Corollary 1. GNNs with max-pooling encodes appropriate non-linear operations, and thus extrapolates under appropriate training sets (Theorem 3).

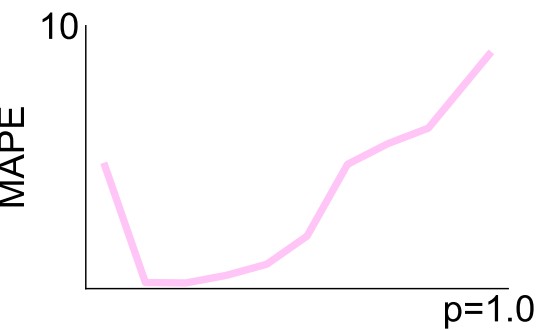

Figure 18: **Shortest path: random graphs.** We train GNNs with neighbor and graph-level min on random graphs with probability $p$ of an edge between any two vertices. x-axis denotes the $p$ for the training set, and y-axis denotes the test/extrapolation error on unseen graphs. The test errors follow a U-shape: errors are high if the training graphs are very sparse (small $p$) or dense (large $p$). The same pattern is obtained if we train on specific graph structure.

