# OpenReview forum: "How Neural Networks Extrapolate: From Feedforward to Graph Neural Networks"
_ICLR.cc/2021/Conference — ICLR 2021 Oral_

### Official Review · AnonReviewer3 · 2020-10-27
**An interesting paper that opens new directions to better extrapolate our current knowledge about deep learning**

**Rating:** 8
**Confidence:** 4

**Review:**

This paper tackles the challenging question of how deep networks might learn to extrapolate knowledge outside the support of their training distribution. The paper contributes both with novel theoretical arguments as well as with empirical evidence collected on targeted cases. Differently from other recent approaches to the problem, the theoretical analyses presented here are non-asymptotic and provide precise information about the kind of functions that MLPs can learn in the proximity of the training region. Moreover, the authors provide compelling arguments about the need to explicitly encoding (task-specific) non-linearities in the input representation and/or in the model architecture in order to promote successful extrapolation.
Overall, the paper addresses important issues and can be considered at the frontier of deep learning research. The paper is well-written and the recent literature is properly reviewed. In light of this, I think the paper would be of interest to the ICLR community. However, I would like to explicitly mention that I was not able to carefully review all the details and proofs reported in the Appendix, which is of an unusual length (almost 40 pages) for an ICLR paper.

Comments for possible improvements:
- The analyses reported in Appendix D.3 / C.4 regarding the extrapolation capability of MLPs with different activation functions (sin, tanh, quadratic) are relevant and should be emphasized. They could also be expanded, for example by considering some data generation tasks analyzed in the main text.
- It would be very interesting to extend this analysis to other simple problems, where MLPs cannot extrapolate appropriately. I am specifically referring to the simple counting and arithmetic tasks discussed in [1], where generalization outside the training distribution was achieved by adding ad-hoc gate units to the network. I think this domain is particularly relevant here, given that arithmetics is mentioned by the authors in the opening sentence of the paper.

[1]	A. Trask, F. Hill, S. Reed, J. Rae, C. Dyer, and P. Blunsom, “Neural Arithmetic Logic Units,” in arXiv:1808.00508, 2018.

---

> ### Author Response · Authors · 2020-11-21
> **Our response**
>
> Thank you for your insightful feedback.
>
> We have added additional experiments on MLPs with tanh, quadratic, and cosine activation functions in Section 3.3. We explore the extrapolation ability of these MLPs on tasks that we used for ReLU MLPs. In general, an MLP is better at extrapolating functions that involve non-linearities “similar” to the MLP’s activation, e.g., quadratic MLP extrapolates well when learning quadratic functions. We leave further theoretical analyses to future work.
>
> Thank you for pointing us to previous works on arithmetic tasks and neural arithmetic logic units (NALUs). They are indeed quite related. In Section 4.1, we use our theoretical results to offer a potential explanation on why NALUs help extrapolation in arithmetic tasks. To learn multiplication, NALUs encode a log-and-exp non-linear transform in the architecture. Since log(a * b) = log a + log b, this transform reduces multiplication to a linear function, which helps extrapolation following our linear algorithmic alignment hypothesis. To improve learning addition operations, they propose sparsity constraints, which is beyond the scope of our paper.
>
> We are happy to answer any other questions you may have.

---

### Official Review · AnonReviewer1 · 2020-10-27
**Important work that enhances our understanding of graph neural networks. Ideas are relevant, solid, and well supported. Excellent work overall.**

**Rating:** 9
**Confidence:** 4

**Review:**

This paper investigates the extrapolation power of MLPs and GNNs (trained by gradient descent with mean squared loss) from a theoretical perspective. The authors show results of extensive experiments that back up their theoretical findings.

In particular, the authors study the question of what these neural networks learn outside the training distribution, and identify conditions when they extrapolate well. Their findings suggest that ReLU MLPs extrapolate well in linear tasks, with a fast convergence rate (O(1/\epsilon). GNNs (having MLP modules) extrapolate well when the non-linear operations are encoded in either network architecture or data representation, so as the inner MLP modules are aligned with only linear functions.

The paper is well written, ideas and definitions clearly explained and experiments laid out in detail. The theoretical contributions of the work are important as they enhance our understanding of how these networks learn and how well they generalize. These findings help us design GNNs based on the data and problem at hand. As such, this work addresses a fundamental question in GNN understanding and must be published.

Some comments/questions to the authors:
- In Section 3.2, “diversity” of a distribution is informally defined in terms of training support and direction. A more thorough definition would be helpful.
- The title of the paper is somewhat misleading: “from feedforward to GNN” insinuates that there are other network types that are discussed in the paper.

---

> ### Author Response · Authors · 2020-11-21
> **Our response**
>
> Thank you for your insightful feedback. We answer your questions below.
>
> Q1: In Section 3.2, “diversity” of a distribution is informally defined in terms of training support and direction. A more thorough definition would be helpful.
>
> A1: We have provided an exact definition of “diversity” in Theorem 5. By “direction”, we are referring to the non-zero vector w.
>
> Q2: The title of the paper is somewhat misleading: “from feedforward to GNN” insinuates that there are other network types that are discussed in the paper.
>
> A2: We are sorry for your confusion. The title refers to the relation that GNNs are built on feedforward network modules. In our paper, our analysis of feedforward networks (Section 3) leads to our understanding of the more complex GNNs (Section 4). We hope the title is appropriate from this perspective. Please let us know if you still have concerns.
>
> We are happy to answer any other questions you may have.

---

### Official Review · AnonReviewer4 · 2020-10-27
**Crucial study on extrapolate ability of MLP and GNN that provides a different aspect of analysis on multi-domain adapatation**

**Rating:** 9
**Confidence:** 4

**Review:**

This paper analyzes the extrapolate ability of MLPs and GNNs. In contrast to the existing theoretical works that focus on generalizability  and capacity of these models, this paper emphasizes the behavior of training algorithm using gradient descent. It takes analogy of kernel regression via the neural tangent kernel as an example to study the bias induced by the gradient descent algorithm. The presentation of this paper is clear and well-organized with the most significant result shown in the first section, raising interest of the readers, as opposed to leaving them behind a massive amount of proofs. The contribution of this paper is significant as well since it draws attention of the researcher to theoretical analysis on the bias induced from the implementations of the algorithms as compared to the theoretical analysis on the model structure itself. Model extrapolation is also closely connected to topics such as meta-learning, multi-task learning, domain adaptation and semi-supervised learning since the ability of model extrapolation will limit its performance when applied to other tasks.

Pros:
1. This paper has shown some interesting results: for instance, MLP with ReLU trained by GD will converge to linear functions along any direction from origin outside the support of the training data. This coincide with the idea that MLP are piecewise linear in different regions. The proof is complicated though and requires the analogy to the kernel regression as basis.  This result seems to suggest that the learning of MLP on data manifold supported by training data is also local linear and without support of training data, the induction follows the inertia of linearity.  It is curious to see if this is due to the piecewise linearity of ReLU function.  Maybe we will have better nonlinear extrapolation for MLP using tanh and other sigmoid functions.
2.  Comparison between GNN and Dynamic programming algorithm is very intuitive and inspiring. It suggests that max/min aggregate as opposed to more commonly used sum-aggregate in GNN is more suitable for extrapolation and the similarity between max/min aggregate GNN and DP is also very convincing. In general, this paper built up a good intuition before diving into the proof, which is well-appreciated.
3.   The suggestion to improve extrapolation is to put the nonlinearity into the architecture of the GNN or into the input representation is useful. For instance, replacing sum-aggregate to min/max aggregate helps to achieve good extrapolation. It also explains why the pre-trained embeddings such as BERT can be used in other tasks and still extrapolate well.


Suggestions:
1. Limitations of the study scope. This paper only discuss results of neural network using ReLU and GD. Although GD is widely used, the ReLU as the activation function plays a critical role in the study of extrapolation. It is necessary to provide analysis on the use of other common activation function to understand if the extrapolation ability is expanded.
2.  It is interesting to see more connection with domain adaptation and semi-supervised learning as well.

---

> ### Author Response · Authors · 2020-11-21
> **Our response**
>
> Thank you for your insightful feedback.
>
> We have added additional experiments on MLPs with tanh, quadratic, and cosine activation functions in Section 3.3. We explore the extrapolation ability of these MLPs on tasks that we used for ReLU MLPs. In general, an MLP is better at extrapolating functions that involve non-linearities “similar” to the MLP’s activation, e.g., quadratic MLP extrapolates well when learning quadratic functions. We leave theoretical analysis to future work.
>
> We have added Section 5 to discuss the connections of our results with other out-of-distribution settings including domain adaptation, self-supervised learning, invariant models, and distributional robustness. We conjecture that some of these methods may improve extrapolation by (1) learning useful non-linearities beyond the training data range from unlabeled out-of-distribution data and (2) mapping relevant out-of-distribution test data to the training data range.
>
> We are happy to answer any other questions you may have.

---

### Official Review · AnonReviewer2 · 2020-10-28
**Interesting paper with somewhat specific results**

**Rating:** 9
**Confidence:** 3

**Review:**

## Summary

The paper studies how neural networks extrapolate. The authors theoretically
examine two-layer ReLU MLPs with mean squared loss in the NTK regime and,
building on these results, GNNs. They find that the MLPs quickly converge to
linear functions along any direction from the origin, but can provably learn a
linear target function where the training distribution is sufficiently diverse.
For GNNs, they propose a hypothesis that the success of extrapolating
algorithmic tasks to new data relies on encoding task-specific non-linearities
in the architecture or features. The theoretical results are supported by
empirical results which sometimes go beyond the specific conditions of the
theorems (like increasing the number of layers in the MLP to 4 in Appendix
C.1.).

## Pros

- The paper provides both theoretical and practical insight into the
  extrapolation capabilities of neural networks, especially GNNs.
- I especially liked the part about GNNs and the hypothesis that if we can
  encode the non-linearities outside the MLPs so the MLPs only have to learn
  linear functions, GNNs will extrapolate well.
- Overall I found the paper very interesting and fun to read.

## Concerns

- The theoretical MLP results are very specific. Sometimes this is not apparent
  from either the abstract or the discussion of the results. Some of the
  constraints:
  + The MLPs have two layers, which I find the most limiting constraint as most
    practical MLPs have more layers.
  + The mean squared loss is used throughout the paper. I think this is not
    emphasized enough (it is mentioned only a single time in the paper). As far
    as I understand the proofs also rely on the loss, so the loss should be
    included in the conditions of the theorems.
  + We are under the NTK regime, which is of course evident from the techniques
    used. Nevertheless, this is not mentioned in the abstract.
  + The MLPs are ReLU MLPs which is emphasized sufficiently in the paper. The
    authors include preliminary empirical results for other activations
    functions in the Appendix (sin, quadratic, and tanh).

## Questions

- Could the proofs of Theorem 3 and Theorem 5 be generalized to MLPs with more
layers?
- Can we gain some insight into extrapolation with other loss functions like
softmax based on these results?

## Reasons for ranking

I found the paper really interesting and gained much insight from it. Some of
the constraints of the MLPs are not emphasized enough and the writing is more
general at parts than the results warrant. Even with the constraints I believe
that this is an important step and sheds light on the extrapolation capabilities
of neural networks. If the constraints can be made clearer I'm willing to
improve my score further.

## Minor comments

- Second to last paragraph on page 5: "for Theoreom 5" should be "for Theorem 5".
- Caption of Figure 1: outisde => outside
- In 4.2., "Experiments: architectures that help extrapolation": "GNNs with
  max-readout are better than GNNs with sum-readout (Fig 6a)" should be Fig 5a.

---

> ### Author Response · Authors · 2020-11-21
> **Our response**
>
> Thank you for your insightful feedback.
>
> We have made the assumptions of our theorems clearer throughout the paper: (1) In the abstract, we now state that our theoretical results build on the connection between overparameterized networks and the neural tangent kernel; (2) We have clarified that we are using the squared loss in all theorems; (3) In the introduction and Section 3, we have emphasized that our proofs are for two-layer networks. As you have realized, we use experiments to confirm that our theory holds across different training settings, such as 4-layer networks (Appendix C.1 and C.2). Therefore, the assumptions in the theorems can be relaxed in practice. Thank you again for your helpful suggestions. Please let us know if any part remains inaccurate, and we will fix it in the final version.
>
> We answer your questions below.
>
> Q1: Could the proofs of Theorem 3 and Theorem 5 be generalized to MLPs with more layers?
>
> A1: After some preliminary calculation, we believe our proof techniques can be extended to more than two layers. However, a full proof for more layers requires significant effort, so we do not have a complete proof at the moment. Note that most theoretical works on NTK focus on two layers for similar reasons. As you may have noticed, we do have experimental results to confirm that our theory holds for deeper networks (Appendix C.1 and C.2). We agree that extending the proof to multiple layers is an important future direction.
>
> Q2: Can we gain some insight into extrapolation with other loss functions like softmax based on these results?
>
> A2: Unfortunately, it is hard to extend the theory to the softmax loss, but, we agree this is an important direction for future work. Note that the squared loss can be competitive to other losses for classification [1], so the lessons we learned from regression tasks may also be useful for classification tasks.
>
> [1] On Loss Functions for Deep Neural Networks in Classification. Janocha et al. 2017
>
> We have fixed the grammar mistakes as suggested.
>
> We are happy to answer any other questions you may have.

---

> > ### Comment · AnonReviewer2 · 2020-11-21
> > **Response**
> >
> > Thank you for your detailed response!
> > I really like the paper and my concerns were addressed so I updated the score to 9.

---

> > > ### Author Response · Authors · 2020-11-21
> > > **Thank you**
> > >
> > > Thank you! We are glad you like our paper, and we appreciate your insightful comments.

---

### Author Response · Authors · 2020-11-14
**General Update**

Dear Reviewers and AC,

We sincerely appreciate all the reviews. They give positive and high-quality comments on our paper with a lot of constructive feedback. We are working on incorporating the insightful and valuable suggestions from the reviewers. We will update the draft and post the response soon.

---

### Author Response · Authors · 2020-11-21
**Update**

Dear Reviewers and AC,


We have updated our draft to incorporate the insightful suggestions of the reviewers:

Following Reviewer 3 and  Reviewer 4’s suggestion, we have added additional extrapolation experiments on MLPs with different activation functions (tanh, quadratic, and cosine) in Section 3.3 (preliminary results previously presented in Appendix).

Following Reviewer 4’s suggestion, we have added a section (Section 5) on relations to other out-of-distribution settings, including domain adaptation, self-supervised learning, invariant models, and distributional robustness.

Following Reviewer 2’s suggestion, we have made the assumptions of our theorems clearer throughout the paper. We have also emphasized that our theoretical results empirically hold across different training settings (e.g., width, depth, learning rate, batch size), so the assumptions may be relaxed in practice.

Following reviewer 3’s suggestion, we have discussed the related work Neural Arithmetic Logic Units in Section 4.1. Our results may suggest an explanation why their proposed architecture improves extrapolation in arithmetic tasks.

We will improve other minor points of Reviewer 1, Reviewer 2, Reviewer 3, Reviewer 4 in the final version. Thank you all for the valuable suggestions.

Please let us know if you have additional questions.


Thank you,

Authors

---

### Decision · Program_Chairs · 2021-01-07
**Final Decision**

**Decision:**

Accept (Oral)

**Comment:**

This paper studies how (two layer) neural nets extrapolates. The paper is beautifully written and the authors very successfully answered all the questions. They managed to update the paper, clarify the assumptions and add additional experiments.